# Modelling temporal variability of *in-situ* soil water and vegetation isotopes reveals ecohydrological couplings in a riparian willow plot

Aaron Smith[1], Doerthe Tetzlaff[1,2,3], Jessica Landgraf[1], Maren Dubbert[1,4], Chris Soulsby[3,2,5]

[1]IGB Leibniz Institute of Freshwater Ecology and Inland Fisheries Berlin, Berlin, Germany
[2]Humboldt University Berlin, Berlin, Germany
[3]Northern Rivers Institute, School of Geosciences, University of Aberdeen, UK
[4]ZALF Leibniz Center for Agricultural Landscape Research, Müncheberg, Germany
[5]Technische Universitat Berlin, Chair of Water Resources and Hydrosystems Modelling, Berlin, Germany

*Correspondence to*: Aaron Smith (smith@igb-berlin.de)

**Abstract.** The partitioning of water fluxes in the critical zone is of great interest due to the implications for understanding water cycling and quantifying water availability for various ecosystem services. We used the tracer-aided ecohydrological model EcH2O-iso to use stable water isotopes to help evaluate water, energy, and biomass dynamics at an intensively monitored study plot under two willow trees, a riparian species, in Berlin, Germany. Importantly, we assessed the value of *in-situ* soil and plant water isotope data to help quantify xylem water sources and transit times, with coupled estimates of the temporal
dynamics and ages of soil and root-uptake water. The willows showed high water use through evapotranspiration, with limited percolation of summer precipitation to deeper soil layers due to the dominance of shallow root-uptake (>80% in the upper 10cm). Lower evapotranspiration under grass resulted in higher soil moisture storage, greater soil evaporation and more percolation of soil water. Biomass allocation was predominantly foliage growth (57% in grass and 78% in willow). Shallow soil water age under grass was estimated to be similar to under willows (15-17 days). Considering potential xylem transit times
showed a substantial improvement in the model's capability to simulate xylem isotopic composition and water ages; and demonstrate the potential value of using *in-situ* data to aid ecohydrological modelling. Root-uptake was predominately derived from summer precipitation events (56%) and had an average age of 35 days, with xylem transport times taking at least 6.2 – 8.1 days. By evaluating isotope mass balances along with water partitioning, energy budgets and biomass allocation, the EcH2O-iso model proved a useful tool for assessing water cycling within the critical zone at high temporal resolution,
particularly xylem water sources and transport; which are all necessary for short and long-term assessment of water availability for plant growth.

## 1 Introduction

Understanding how water is partitioned in the Critical Zone (CZ), the near-surface zone from the top of the vegetation canopy to groundwater (Grant and Dietrich, 2017), is essential for improving knowledge of landscape functionality while providing
an evidence base for sustainable water management strategies. The partitioning of CZ water is strongly dependent on

evapotranspiration (ET) (which accounts for >60% of terrestrial precipitation) (Oki and Kanae, 2006; Zhang et al., 2016), with vegetation water use globally accounting for 65-70% of evapotranspiration (Good et al., 2015; Schlesinger and Jasechko, 2014). While measurements of ET and transpiration fluxes help to quantify water partitioning, these measurements usually do not constrain the dynamics of how water is taken from different water sources (i.e. different soil depths) which may greatly

change with wetness conditions (Rothfuss and Javaux, 2017), climate zone (Amin et al., 2020), and seasonally (Barbeta and Peñuelas, 2017). In regions where ET dominates precipitation water partitioning (>90%, Zink et al., 2017), seasonal variations in ET greatly reduce water availability during the growing season. Additionally, the importance of evaluating the water footprint of biomass production in water-limited regions where ET is dominant highlights the importance of partitioning ET into its components of interception and soil evaporation and transpiration (Kool et al., 2014; Xiao et al., 2018). However,

partitioning ET to quantify the transpiration component throughout the growing season is complicated by multiple factors; including atmospheric demand, vegetation and root conductance, rooting distribution, stomatal resistance, and water potential throughout the soil profile (Dubbert and Werner, 2019; Jones and Tardieu, 1998; Sperry and Love, 2015). While total transpiration water usage may be constrained by direct measurement at plot or stand scales (e.g. via sap flow and eddy covariance, Kool et al. (2014)) the source of transpired water from the rooting zone throughout the growing season is not easily

measured and remains highly uncertain (Brantley et al., 2017; Dubbert and Werner, 2019).

Continued efforts to close the knowledge gap on root-uptake sources are essential as increasing climatic variability and accentuated extremes will likely affect future crop and timber production (Lobell and Gourdji, 2012; Lobell et al., 2011) as well as groundwater and stream water availability (Gudmundsson et al., 2019; Taylor et al., 2013). Tracers in soils and xylem, specifically conservative water stable isotopes deuterium ($^2$H) and oxygen-18 ($^{18}$O), have previously been shown to be effective

tools to help constrain root-water uptake sources using various approaches (Rothfuss and Javaux, 2017). These approaches usually use mixing relationships; including linear mixing models of water derived from different pools (e.g. (Barbeta and Peñuelas, 2017)), the similarity of potential source waters and xylem water (e.g. ellipsoid method) (Amin et al., 2020; Tetzlaff et al., 2021), Bayesian mixing frameworks (cf. von Freyberg et al. (2020)), and physically-based modelling approaches (Knighton et al., 2020; Ogle et al., 2014; Sutanto et al., 2012). However, many of these previous studies have utilized data

obtained from destructive sampling, which limits the number of samples and may not account for the transport lag of soil to xylem sample locations (von Freyberg et al., 2020), diurnal variability (De Deurwaerder et al., 2020), or stem internal storage and exchange with xylem (Steppe et al., 2006). Moreover, increasing uncertainty surrounds whether common destructive methods, which involve cryogenic extraction, actually represent xylem water (Chen et al., 2020). Recent developments of *in-situ* tracer measurements of soil and xylem (Marshall et al., 2020; Oerter and Bowen, 2017) have enhanced the possibilities

for spatio-temporal evaluation of root-uptake distributions, in particular when integrated with other data in physically-based ecohydrological models. Higher resolution sampling can be beneficial as it can provide a wider range of temporal conditions (e.g. event rewetting) which helps to test model performance and in the assessment of model structure.

Physically-based ecohydrological modelling approaches have a wide range of applicability, including the estimation of water and energy fluxes and storages, flux partitioning, and biomass accumulation (Asbjornsen et al., 2011; Fatichi et al., 2012;

Maneta and Silverman, 2013). Recent developments linking tracers into ecohydrological models (e.g. Kuppel et al. (2018a)) have further expanded this potential to track water flow paths and associated ages, which can aid in understanding water cycling and mixing within the CZ (Geris et al., 2017; Penna et al., 2018; Sprenger et al., 2019). Such modelling approaches can help overcome the limitations of using isotopic data alone regarding temporal frequency and spatial heterogeneity which may affect estimates of water partitioning and source identification (Goldsmith et al., 2018; Rothfuss and Javaux, 2017;

Sprenger and Allen, 2020). However, relatively few studies have utilized physically-based models to estimate storage-flux-age dynamics while considering mixing of water after root-uptake (Knighton et al., 2020), particularly at high resolution, to account for sub-daily variability (e.g. De Deurwaerder et al. (2020)) or with consideration of root length properties (Gessler et al., 2021; Seeger and Weiler, 2021).

    Here, we utilized soil and xylem water isotope data from *in-situ* monitoring over the 2020 growing season in Berlin, Germany,

to help calibrate a tracer-aided ecohydrological model and estimate water flux and uptake dynamics. The *in-situ* location, which, crucially, also monitored complementary soil moisture and vegetation growth dynamics, comprises a small stand of riparian willow trees and surrounding grassland in a situation typical of NE Germany. The model, EcH$_2$O-iso, is a distributed physically-based model that couples water and energy fluxes, with vegetation carbon allocation across the soil-plant-atmosphere continuum (SPAC). The primary goal of this study was to utilize the EcH$_2$O-iso model as a learning framework

to evaluate the temporal linkages of energy, water and ecology of the plot site throughout the growing season. The primary research goal was evaluated through the following research questions: 1) how well can the tracer-aided ecohydrological model reproduce *in-situ* measurements of fluxes, storages, and stable water isotopes of two juxtaposed vegetation types (willows and grass); 2) does distance-based isotopic mixing of root-water uptake better approximate xylem water isotopes than uniform instant root-uptake mixing; and 3) what are the transit times of vegetation water and what are the implications for water usage.

We used multicriteria calibration of water, energy, biomass, and isotopic datasets as equally informative data within the EcH$_2$O-iso model, with hydrological, isotopic, and ecological model outputs utilized as tools to evaluate these research questions. Such overall assessment of ecohydrological partitioning, in fluxes as well as root-uptake, was intended to help to improve the conceptualization of water cycling in the CZ at high-temporal resolutions and contribute to an evidence base for management strategies in water sensitive areas.

## 2 Study site and background

### 2.1 Study site

    The study site is in a peri-urban area in the grounds of the Leibniz-Institute of Freshwater Ecology and Inland Fisheries (IGB) in south-east Berlin (Fig 1b). The site is situated near Lake Muggelsee (80m north of the lake edge), encircled in the North and West by buildings (40m), and east and south (30 and 20m, respectively) by additional vegetation. A stream draining nearby

ponds fringes the East of the study area. A lake water extraction facility (using bank filtration) is situated immediately north of the IGB site and groundwater is ~2.2m below the ground surface with limited annual variability (<0.1m). A previous isotope-

based study at the site has excluded groundwater and the nearby stream water as likely sources of water to the trees (Landgraf et al., 2021).

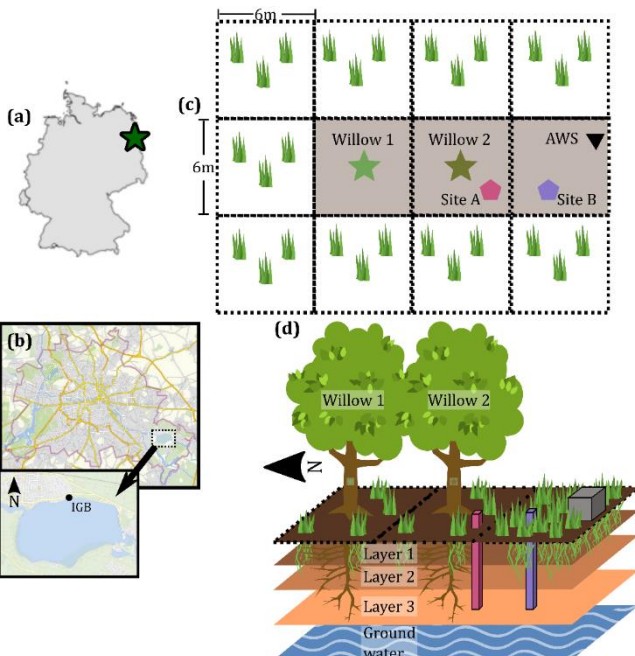

**Figure 1: Study site location and site description. (a) Location of the site within Germany, (b) Location of IGB within Berlin (Berlin, 2021), (c) plan view of the study site including the location of willow trees, soil water isotope and moisture, and automatic weather station (AWS), and (d) conceptual diagram of the study site, with three soil depths and two primary vegetation units (with relative prominence). Pink and blue columns in subplot d indicate soil moisture and isotope measurement columns, and grey box indicates AWS location. Berlin Maps © OpenStreetMap contributors 2021. Distributed under the Open Data Commons Open Database License (ODbL) v1.0. The area surrounding the study site is grass-covered, similar to Site B.**

## 2.2 Soils and vegetation

The study site was chosen based on its open nature (not shaded) for the two willow trees (*Salix alba*) and an adjacent automatic weather station (Fig 1c&d). Willows are a riparian species, which is well adapted to moderate and high moisture typical of riparian areas or with high groundwater levels (Isebrands and Richardson, 2014). Additional site selection criteria are provided in Landgraf et al. (2021). The willows have a similar age of ~15 years with willow in the North (Willow 1, Fig 1c&d) being slightly larger (~9m height and stem diameter: 398mm, 16.07.2020) and the southern willow (Willow 2, Fig 1c&d) being slightly smaller (~8m height and stem diameter: 353mm, 16.07.2020). Sparse grass is present below the willow trees, with some bare soil patches (Fig. 1d). Grass coverage is greater with no clear bare soil patches in the open area surrounding the willows (Fig. 1d).

The site is situated in the North European Plain, with geology and surficial soils in the surrounding areas of Berlin deposited during the Weichselian glaciation (Deutschen Nationalkomitee, 2016). The willows are situated on reclaimed ground where a previous building was demolished and the site backfilled with sandy brown earth topsoil. Soil samples taken to 1m depth reveal

a relatively uniform soil structure, with a slightly higher organic horizon in the near-surface soils where rooting is densest, which is more developed below the open grassland area south of the willows (Site B, Fig. 1c&d).

## 2.3 Climate

The climate is continental with a maritime influence (Köppen Index Cfb) and experiences substantial interannual variability in precipitation (Table 1). The range of annual precipitation from within the climate normals (1980-2010) is 80% of the long term average precipitation (526 mm/year, Table 1) (DWD, 2021). While precipitation during the study period was similar to the climatic normal, the site experienced lower than normal humidity, and higher wind speeds and air temperature (Table 1). The growing season is less humid and is warmer than annual averages (Table 1), with large sub-daily variability hydroclimate. For the vegetation growing season in the surrounding region as well as Berlin, evapotranspiration (ET) is the dominant hydrologic flux, accounting for ~90% of total precipitation (Gillefalk et al., 2021; Smith et al., 2020a).

**Table 1: Climate conditions at a nearby long-term weather station, Berlin-Brandenburg airport (DWD, 2021). Average (standard deviation in parentheses) climate conditions are shown for the climate normals (1980 - 2010) and the study year (2020). The growing season is May – September (inclusive). NA is not applicable.**

| | Climate Normals (1980 -2010) | Study Year (2020) | Study Growing Season (2020) | Daily Average High (Low) in Growing Season (2020) |
|---|---|---|---|---|
| Precipitation (mm/year) | 526.5 (104.5) | 529.1 | 257.1 | NA |
| Relative Humidity (%) | 76.8 (2.1) | 71.9 | 64.2 | 90.5 (42.9) |
| Wind Speed (2m, m/s) | 3.10 (0.2) | 3.5 | 2.9 | 5.0 (1.0) |
| Air Temperature (2m, °C) | 9.3 (0.9) | 11.4 | 17.8 | 23.5 (11.8) |

## 3 Materials and methods

### 3.1 *In-situ* measurements

The study site was set up to continuously measure hydroclimate, soil moisture, and vegetation productivity. Hydrometeorological conditions at the site were monitored with a mobile Eddy Covariance system (Li-cor Biosciences, Lincoln, NE, USA), which included measurements of precipitation amounts (event), collection of precipitation samples for isotope analysis, humidity, air temperature, wind speed, short wave radiation, surface temperature, and latent heat (Site B, Fig 1c). Climate data were quality checked against nearby weather stations that have been in operation longer. These include measurements from the roof of the nearby IGB building (2013), the station on Lake Muggelsee (2013), and surrounding DWD weather stations (DWD, 2021). Precipitation samples were collected at 4-hourly intervals. Each sample bottle was filled with 0.5cm of paraffin oil to prevent evaporation. Precipitation samples were analysed with a Picarro L2130-i cavity ring-down laser spectrometer (Picarro, Inc., Santa Clara, CA, USA). Hourly pictures were taken from the soil surface towards the canopy to capture changes in leaf coverage. Images were translated to an estimated leaf area index (LAI) using the LAI package for R (Martin, 2015) which estimates the gap fraction of the histogram-based unimodal threshold method.

Soil moisture, isotopes ($\delta^2H$ and $\delta^{18}O$), and temperature were measured at two sites in the study area, Site A near Willow 2 and Site B in the open grass area. Soil moisture and temperature were measured at three depths (10, 40, and 100cm) with water content reflectometers (CS616, Campbell Scientific, Inc. Logan, UT, USA. ±2.5% for VMC) and thermistors (BetaTherm 100K6A1IA, Campbell Scientific, Inc. Logan, UT, USA. ± 0.2ºC). Visual inspection of rooting densities during installation suggested a higher proportion of roots near the surface and very few tree roots at 1m.

Soil water isotope samples were collected using both destructive bulk soil sampling and with an *in-situ* vapour isotope analyser. Installation of polypropylene membranes (7cm long, 0.2μm pores, Kübert et al., 2020) for soil water vapour extraction at three depths (10, 40, and 100cm) was conducted at the end of May 2020 (May 20, 2020). Extracted *in-situ* soil vapour samples were analysed every two hours using a Picarro L2130-I cavity ring-down laser spectrometer (Picarro, Inc., Santa Clara, CA, USA), calibrated using the approach presented for this site in Landgraf et al. (2021).

To complement the climate and soil measurements, detailed measurements indicative of vegetation water fluxes and biomass accumulation were conducted continuously throughout the study period. Measurements included sap flow (heat ratio method, SFM1 instrument, ICT International, Australia), dendrometers (DR, Radius Dendrometer, Ecomaticl, Dachau, Germany), and xylem water vapour (Picarro, Inc., Santa Clara, CA, USA). Polypropylene membranes (as per the soil) were installed in two bore holes in both willow trees starting in June (Willow 2). COVID-19 related complications delayed the installation in Willow

1 (August). Xylem water vapour was extracted from two bore holes (inner diameter 8-10mm, described in Marshall et al., 2020) on each willow tree, at 30cm and 170cm, and analysed in the same analyser as for the soil vapour isotopes every other hour. To prevent contamination of atmospheric air within the xylem vapour samples, the probe was embedded within the stem and sealed using commercial silicon. Analyzed xylem water isotopes were evaluated for wounding effects (in early periods after installation) and any were removed from the analysis. Wounding effects were identified by daily inspection of the trees

and unrealistically enriched isotopic compositions which exponentially decreased in time.

Insulated heated cables were installed with the tubing for all vapour to avoid condensation effects and modulate vapour temperatures. Additionally, tubing was flushed with dry air for 10 minutes each morning if condensation was detected to remove any residual water. To test and correct for water concentration and temperature dependencies, a linear regression of vapour concentration and slopes of $^2H$ and $^{18}O$, and nonlinear polynomial regression of daily average xylem and soil water

isotopes with temperature (air and soil, respectively) was conducted to determine the strength of the relationship for water concentration and temperature-dependent offset. Corrected *in-situ* xylem and soil water isotopes for water concentration and temperature were consistent with bulk soil water isotope and twig cryogenically extracted isotope samples (see Supplementary Material and Landgraf et al. (2021) for more details on isotopic measurement).

### 3.2 EcH₂O-iso model

EcH₂O-iso is a physically-based, distributed, tracer-aided model coupling vegetation, and soil energy and water balance, and carbon utilization (Maneta and Silverman, 2013). The isotope module (simulating $\delta^2H$, $\delta^{18}O$, and water age) was coupled with the water balance to track the movement of water throughout the model domain (Kuppel et al., 2018a). The following section

presents a brief synopsis of the model conceptualisation of energy, water, and isotope balances (conceptual diagram, Fig. S1). Further details of the model are provided in Maneta and Silverman (2013) and Kuppel et al. (2018a).

### 3.2.1 EcH₂O-iso energy balance

The energy balance is resolved with a top-down approach, conducting energy balance within the canopy before the energy balance at the surface. The canopy energy balance partitions incoming radiation into latent heat (interception evaporation and transpiration), sensible heat, and net radiation as a function of canopy temperature (Fig. S1). The canopy energy balance is very sensitive to the canopy stored water (maximum canopy storage parameter, $CWS_{max}$), where higher intercepted water storage decreases energy availability for transpiration. Transpiration is limited by environmental constraints, implemented using a Jarvis-type stomatal conductance model dependent on soil moisture, vapour pressure deficit, air temperature, and incoming radiation. EcH₂O-iso was modified to account for potential root-uptake from outside the vegetation model cells. Radial rooting parameterisation (Section 3.3.1) estimated the rooting proportion in each surrounding model cell which assumes a radial distance from the cell centre. Rooting proportions with depth ($k_{root}$, described in Kuppel et al., 2018) and laterally were utilized to account for total transpiration water availability and lateral proportions used to withdraw water from neighbouring cells.

The surface energy balance utilizes energy translated from the canopy energy balance to resolve latent heat (soil evaporation), sensible heat, net radiation, ground heat, and snow and melt heat (only if snow is present) using surface temperature. Ground heat is resolved using two thermal layers, where the depth of the first thermal layer is defined as half of the depth where the thermal wave is damped by 37% (Arya, 2001). Interpolation of soil temperatures for each soil layer are estimated using surface and soil temperature at the bottom of the thermal layers, and the linear damping equation (Arya, 2001) solved at the bottom of each soil layer. Soil temperature interpolation does not influence the energy balance.

### 3.2.2 EcH₂O-iso water balance

Similar to the energy balance, the water balance is estimated with a top-down approach including canopy, surface, and sub-surface storage (Fig. S1). Canopy storage is estimated using a linear bucket approach, with maximum storage limited by a storage parameter ($CWS_{max}$) and canopy interception driving draw-down of canopy storage. Net precipitation as throughfall accumulates on the soil surface (ponded water) and infiltrates into the shallow soil (layer 1) using the Green-Ampt model (Brooks-Corey, air-entry pressure, and vertical hydraulic conductivity parameters, $\lambda_{BC}$, $\psi_{ae}$, and $K_v$, respectively). Ponded water at the end of each time-step is directly routed to ponded water in the next downstream cell. Soil water redistribution is conducted using gravitational drainage when field capacity is exceeded. Redistribution is estimated for each model layer. The model assumes that there is no preferential flow in any soil layer. Very dry soils at the study site prompted the introduction of sub-discretization of shallow soils (layer 1) to estimate the moisture at discrete depths in addition to the average moisture of the layer. Shallow soils were sub-discretized into 1cm increments with incoming water (infiltration and return flow) entering from the layer boundaries and redistributed using gravitational drainage. Sub-discretization was implemented for informational

purposes only, not for use in calibration. Gravitational drainage rates linearly increase from zero (at field capacity) to saturated vertical conductivity when the soil is fully saturated. Upward redistribution of water occurs if deeper soil storages are fully saturated. Vertical downward flux from the deepest soil layer (layer 3) can occur due to leakance out of the model domain. Lateral flow may occur in the deepest soil layer, with water above field capacity routed to the next downstream model cell using a linear kinematic wave model.

### 3.2.3 EcH$_2$O-iso isotope mixing and water ages


Mixing of isotopes ($\delta^2$H and $\delta^{18}$O) and water age is conducted using a complete mixing assumption (fully mixed at the end of each time-step) in each soil storage (layers 1-3 independently) and in the canopy and ponded water. Storage mixing is conducted with amount weighted-averaging of isotopic compositions (or age) with incoming fluxes. Out-fluxes have the same isotopic composition (and age) as the mixed water in storage. Evaporative fractionation is calculated for the shallow soils

(layer 1) using estimated soil evaporation and the Craig-Gordon fractionation model (Craig and Gordon, 1965). Humidity in the soil is estimated using the method proposed by Lee and Pielke (1992), with the kinetic fractionation factor modified for use in soils (Braud et al., 2005). Isotopic composition and age of transpired water are estimated by amount weighting the contribution of root-uptake from each soil layer (root-uptake proportion dependent on water availability and rooting distribution). Amount-weighted root-uptake assumes instant translocation, complete mixing, and uniform isotopic composition

throughout the vegetation using soil water isotopes of the same time-step. At the end of each time-step the age of water in storage is advanced by one time-step (e.g. 1 hour).

To evaluate the distribution of water ages, parameter sets were re-run for each time-step that included precipitation (925 hours with precipitation over the study period). The input precipitation concentration was changed in a stepwise manner with a concentration of "1" on the precipitation event evaluated (Smith et al., 2020). For example, the 25[th] precipitation event (i.e.

25[th] hour with precipitation, here, January 7[th], 11:00am), the precipitation concentration was set to 1 while all other precipitation and initial storages had a concentration of 0. The concentration was tracked through the domain to provide a proportion of water in each storage originating from each precipitation event. The cumulative concentration of different precipitation events sums to 1 when all of the water in storage is younger than the duration of the simulation.

### 3.3 Tree water mixing and transit times

Knighton et al. (2020) showed that improved simulation of xylem isotopes could result from considering potential vegetation storage, and mixing of soil water older than one time-step within EcH$_2$O-iso. Similarly, Seeger and Weiler (2021) showed promising conceptualization of isotope mixing within vegetation based on a convolution approach using *in-situ* measurements of soil and xylem water. To evaluate the dynamics of transit times from the root to measurement height during the study period, a combination of modelled results of sap flow, soil water isotopes, and spatio-temporal root uptake proportions were used to

estimate vegetation xylem water composition and age at the average measurement height (1m) while accounting for spatial variability of soil water isotopes (laterally and with depth). Root-uptake proportions were calibrated using parameterised

exponential distributions for depth and lateral contributions in EcH$_2$O-iso. The tree water mixing routine (applied post-EcH$_2$O-iso calibration) was developed with the assumption that isotopic mixing is dependent on the rooting distance (similar to Seeger and Weiler (2021)). Mixing of source water was conducted using the convolution equation and expanded on Seeger and Weiler

(2021) to include lateral soil heterogeneity (e.g. outside model cell or different soil layers). Contrary to Seeger and Weiler (2021), rooting distributions are not parameterized through mixing and are calibrated through energy and water balance in EcH$_2$O-iso. This approach differs from tree-mixing within EcH$_2$O-iso which instantly mixes uptake-water throughout the tree, and from previous tree mixing approaches in EcH$_2$O-iso (Knighton et al., 2020) by specifically exploring the capabilities of rooting distributions and uptake travel times to describe xylem water isotopes. Hereafter, the new xylem mixing will be referred

to as distance-based mixing, and the method used by EcH$_2$O-iso is referred to as instantaneous mixing.

### 3.3.1 Vertical and radial rooting length

The average vertical and horizontal distances (and total distance) of roots in each layer to the measurement height were estimated and calibrated in EcH$_2$O-iso with a modification of the root depth and radial spread approach used in Sperry et al. (2016a). The parameterization of this approach is in synchrony with the rooting distribution within EcH$_2$O-iso. Following

EcH$_2$O-iso, the rooting distribution (k$_{root}$) was used to determine the proportion of total roots in each soil layer (Kuppel et al., 2018b). Vertical distances ($v$) from the base of the vegetation to the vertical centre of the biomass in each layer were estimated using a log function of rooting proportions modified from Sperry et al., (2016a & b) to account for unequal rooting proportions with layers using cumulative rooting proportions:

$$v(i) = 0.01 * \ln\left(1 - 0.995 * \left(p(i) - \frac{(P(i)-P(i-1))}{2}\right)\right) / \ln \beta \tag{1}$$

Where $i$ is the soil layer, p is the cumulative proportion of roots (from the surface), 0.995 is a coefficient indicating 99.5% of root biomass, and $\beta$ is a rooting distribution parameter. The rooting parameter in Sperry et al. (2016a & b) was estimated using a linear correlation to the EcH$_2$O rooting distribution (k$_{root}$) to enable estimation in EcH$_2$O ($\beta = -0.0089 * k_{root} + 0.9947$). The radial distance from the stem of the vegetation is estimated as a function of the volume of rooting (Sperry et al., 2016a & b):

$$Vol = d_1 * \pi * (D * a)^2 \tag{2}$$

where $Vol$ is the volume of roots in layer 1, $d_1$ is the depth of layer 1, $D$ is the total depth of the soil (equivalent to total root depth), and $a$ is an aspect parameter, controlling radial distance. The radial distance is then estimated using the proportion of roots in the soil layers (modified from Sperry et al., 2016a&b):

$$r(i) = \left(3 * Vol * \frac{p(i)}{d(i)*\pi}\right)^{0.5} \tag{3}$$

where $r(i)$ is the average radial distance in layer $i$, $p$ is the proportion of roots in layer $i$ estimated from k$_{root}$, and $d$ is the depth of layer $i$ (Fig. 2a). The total average distance ($D(i)$) for roots in each layer is the sum of $v(i)$ and $r(i)$. The radial extent of roots in each layer is then used to estimate the proportion of roots within the cell and in adjacent cells.

### 3.3.2 Xylem transit time, and calibration

Since much of the rooting system below trees are not well characterized, including limited information on the proportion or total lengths of fine roots to transport roots are available, and the translation of xylem velocity to fine root velocity, we simplified the conceptualization of root transport for each soil layer into convolution integrals (Rothfuss and Javaux, 2017). We conceptualize the rooting system in each soil layer similar to a large river (primary root transport) network with many tributaries (fine roots) with precipitation input (root uptake volume). In this way, root water velocity is implicitly accounted for (averaged) across different root diameters. Root-uptake volume-weighted (combined in-cell and off-cell as estimated from EcH$_2$O-iso, Fig. 2c) convolution equations (gamma distribution) were used to estimate xylem $\delta^2$H from each soil layer for each hourly time-step (Fig. 2d). Fractionation during uptake and along the flow path to measurement height was assumed to be negligible. The approach introduces a scale parameter ($\beta$, hours) for each soil layer and one shape parameter ($\alpha$) as a function of total root length (four total parameters) (Fig. 2b). This approach assumes that there is no additional effects of cavitation apart from sap velocity decreases (measured or simulated) within the xylem or roots of the plant.

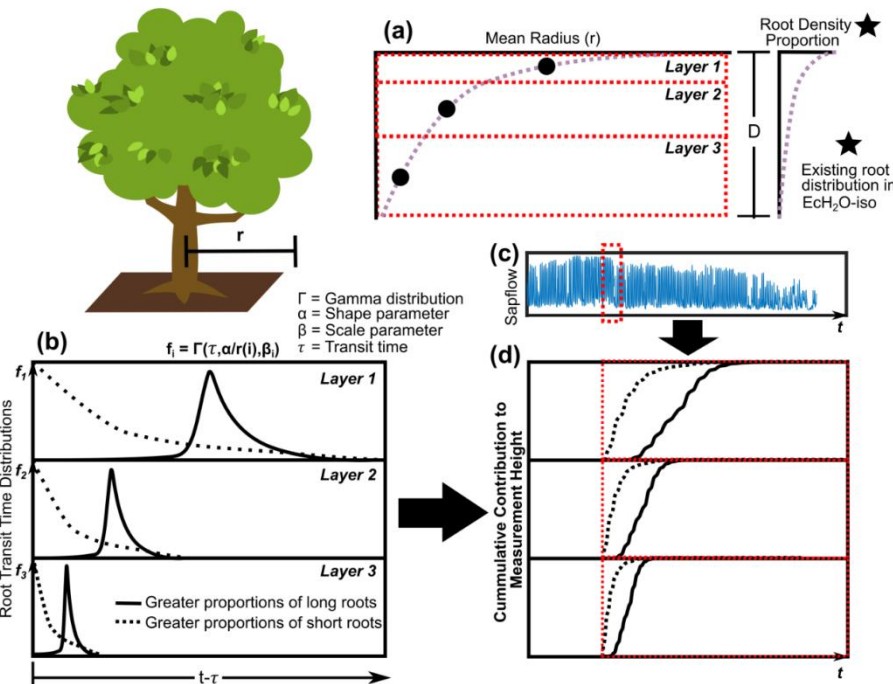

**Figure 2: The estimation of rooting density (a) vertically and (b) the transit time distribution describing root velocity and distance to measurement height weighted by (c) sap flow to get the (d) cumulative contribution of soil water from entry time (red line).**

To evaluate and test the information content of sampling frequency, the convolution integral was calibrated against xylem isotopes at different time-step intervals: using 6, 12, and 24-hour intervals. Measured and simulated xylem isotopes were averaged over the respective intervals prior to evaluation with the Kling-Gupta efficiency (KGE, Kling et al., 2012). While not used in calibration, the KGE of standardized hourly xylem was evaluated to examine how well sub-daily dynamics of soil

water isotopes propagated through the roots. Lastly, to evaluate the effectiveness of distance-based and instantaneous mixing, the transit time and xylem isotopes were calibrated twice, 1) using modelled soil isotopic compositions and sap flow, and 2) using measured soil water isotopes and sap flow. The use of measured soil water isotopes and sap flow tests the maximum potential for how each model performs and is not limited to the performance of $EcH_2O$-iso for sap flow or soil water isotopes. Further, the twin calibration approach shows how the error in the simulation of soil water isotopes potentially propagates through to the estimation of xylem isotope and transit time mixing. Both xylem isotope model calibrations used the root-water uptake proportions estimated from the multi-criteria $EcH_2O$-iso calibration, and are inclusive of off-cell root-water contributions (Section 3.4). The Akaike information criteria (AIC, Akaike (1998)) was used to assess the significance of the additional parameters used by the distance-based mixing (4 additional parameters). The AIC is estimated using the log-likelihood of each model fit and number of parameters, where lower AIC indicates better model performance.

### 3.4 $EcH_2O$-iso and tree mixing model calibration and set-up

### 3.4.1 Model set-up

To best leverage measurements in both willows and the soil pits, the model was set up with three square grids (6m). Two grids contained one willow each and the third contained the open grass area (Fig. 1c). The model was set up to run with hourly time-steps between January 1, 2020, and October 31, 2020, encompassing a spin-up and the primary growing season. Soil layer depths were fixed at the depth of the soil moisture measurements (10, 40, and 100cm) for all model cells. Further, previous analysis and observation during soil moisture probe installation suggest that primary water sources of the willow trees were within the upper 100cm of the soil (Landgraf et al., 2021) which is consistent with shallow rooting of willow trees observed elsewhere (e.g. Philips et al., 2014). Additionally, there is little isotopic evidence of trees using water with distinct groundwater or stream water signatures, despite their proximity (Landgraf et al., 2021). Initial testing revealed inadequate water supply for willows within a single cell (from all soil layers), while measurements revealed no notable decrease in sapflow during the growing season. This suggested potentially substantial water use from outside the measurement location (i.e. model cell), and therefore parameterization of rooting radius (aspect parameter) was set to extend beyond cell boundaries (radius >3m) and permits water use from adjacent cells (assumed to be grass in adjacent cells outside the model domain, Fig. 1c). Increased rooting radius increases the proportion of water used from outside the model cell. Additionally, root distributions with depth were parameterized to be in synchrony with common willow rooting distributions, with a greater proportion of roots near the surface ($K_{root} > 5$). Due to the notable differences in shallow soil moisture between Site A and B (soil organic matter content), soil parameters were different below the willows and the open grass (see calibrated parameters for soil and vegetation, Table S1). Initial soil moisture was set to field capacity which was shown by model testing to have negligible influence on soil moisture simulations during the growing season. Soil water isotopes were initialized using the average measured isotopic composition of soil water. Model calibration of the shallowest soil depths was not sensitive to initial soil water isotope

conditions as simulated isotopes stabilized prior to the beginning of measurement (end of May 2020). The basal diameter of the willows was set to the measured diameter at the start of monitoring.

### 3.4.2 Model forcing data, calibration and evaluation

Hourly model forcing data (Table 2) primarily consisted of data measured with the mobile weather station at the study site (Section 3.1) corrected and gap-filled (where necessary) using the surrounding weather stations (Fig. S2). As long-wave radiation was not measured, estimated long-wave radiation was used from the ERA5 reanalysis dataset (Hersbach et al., 2020) (Table 2). With no direct measurements of LAI in the grass site, MODIS Terra dynamics (MOD15A2, Myneni et al., 2015) dynamics were used to calibrate the LAI. Due to the coarser resolution of MODIS (500m) the LAI range for the grass was scaled to have a maximum of 2 (cf. Smith et al., 2021). In the willow calibration, the MODIS dataset was used in conjunction with canopy images (Section 3.1) in willow LAI calibration.

To best constrain the model performance, and evaluate how well the model estimates the dynamics of water, isotope, and energy fluxes during the growing season, all available data (except $\delta^{18}O$ and soil temperature) were used to constrain model parameterization, with $\delta^{18}O$ retained for further validation (Fig. S3). To simplify the presentation of results, we present $\delta^2H$ as the temporal dynamics of deviation from measurements were not greatly different from $\delta^{18}O$ (Supplementary Material). As the full duration of the single growing season was necessary to constrain fluxes, energy and biomass (e.g. biomass growth and decay) and the model was not used for predictive purposes, a temporal split of the datasets for calibration and validation was not practical. Model results were informally validated using retained datasets ($\delta^{18}O$, sensible heat, and soil temperature) and statistical analyses (root-uptake distribution). To simultaneously evaluate the variability, bias, and correlation of the measured and simulated datasets, the KGE was used for multicriteria calibration of all datasets. Calibration was conducted in two steps, 1) calibration of the isotope, energy, and water balance. and 2) calibration of biomass indices (leaf area index and basal area). To minimize the effects of biomass calibration on energy and water balance, the temporal dynamics of LAI were used as input time series. Second step calibration resampled parameter sets from calibration step 1 for soil and non-biomass vegetation parameters. Each calibration utilized 100,000 Latin Hypercube Sample parameter sets, with parameter combination feasibility evaluated prior to simulations. Infeasible parameter combinations were rejected and resampled from parameter space. An empirical cumulative distribution function (eCDF) was created for each output. For each parameter set, the minimum eCDF value for all variables was used to rank the simulations, retaining the 100 "best" simulations for analysis (Ala Aho et al., 2017). Simulation ranking checked for soil parameter uniqueness to ensure no repeated sampling of soil and non-biomass vegetation parameters occurred within the 100 "best" simulations.

**Table 2: Forcing and calibration data EcH₂O-iso calibration. Parentheses in the calibration datasets indicate the spatial locations of all available measured data.**

| Forcing Data | | Calibration Data | | |
|---|---|---|---|---|
| | Temporal Resolution | | Temporal Resolution | Efficiency |

| Precipitation | Hourly | Soil Moisture (10,40, 100cm; Site A&B) | Hourly | KGE |
|---|---|---|---|---|
| Precipitation Isotopes | 4-hour | Soil $\delta^2$H (10,40, 100cm; Site A&B) | Daily | KGE |
| Air Temperature | Hourly | Surface Temperature (Site B) | Hourly | KGE |
| Relative Humidity | Hourly | Latent Heat (Site B) | Hourly | KGE |
| Wind Speed | Hourly | Scaled Leaf Area Index (Grass) | Daily | KGE |
| Shortwave Radiation | Hourly | Leaf Area Index (Willow 1&2) | Daily | KGE |
| Longwave Radiation | Hourly | Basal Area (Willow 1&2) | Daily | KGE |
| | | Sap flow (Willow 1&2) | Daily | KGE |

## 4 Results

### 4.1 EcH₂O-iso soil water, isotope and energy balance evaluation

Model calibration provided reasonable approximations of the measured variables (Table S2), with the average goodness-of-fit of model simulations within measurement uncertainties (soil moisture MAE range 0.01 – 0.06 m²/m², Table S2). Similar to measured data, simulations of soil moisture and soil water isotopes showed decreasing dynamics with increasing depth (Fig. 3). Despite the close proximity, Site A and B had largely different soil moisture and isotopic response in the upper soil layers (Fig. 3a-d). Simulations were able to capture the drier soil moisture in Site A relative to Site B (Fig. 3a&c); however, the simulated dynamics of average moisture in layer 1 were greater than the measured moisture at 10cm. Sub-discretization of the shallow layer 1 soil (1cm increments using EcH₂O soil water redistribution) using calibrated parameters revealed that while average moisture in the upper 10cm is high (blue lines, Fig. 3a), percolation of infiltrated water down to 10cm during the summer months is limited (Fig. 3a, red line). Late season soil moisture values (average and at 10cm) were over-estimated relative to measurements, coinciding with the underestimation of sap flux during the same period (Fig. 4).

The general dynamics of the soil water isotopes at 10cm in Site A were reasonably represented, with the primary deviation of simulations from measurements due to large sub-daily measured isotope variability (MAE = 8.6‰, Table S2). Additionally, the period of soil water isotopic depletion in August was not captured. Soil moisture and isotope dynamics in the deeper soils (40cm) were much more damped, with a slight under-estimation bias at 40cm at Site B despite appropriate dynamics (Fig. 3d). The much lower variability of isotopes and soil moisture at 100cm was similarly captured by the model calibration. The higher depletion simulated at Site A relative to the measured values was consistent with the soil water isotopic values at greater depth (100cm) measured at Site B. Outliers of *in-situ* soil water isotopes towards the end of the growing season (end of August and September, Fig. 3c-f) were a result of rapid temperature changes which were too strong for the system (i.e. heated cables) to control and caused temporary condensation effects in the tubing. These data were few and did not affect the soil water isotope modelling and uncalibrated soil water isotopic simulations ($\delta^{18}$O, Fig. S3) were broadly similar.

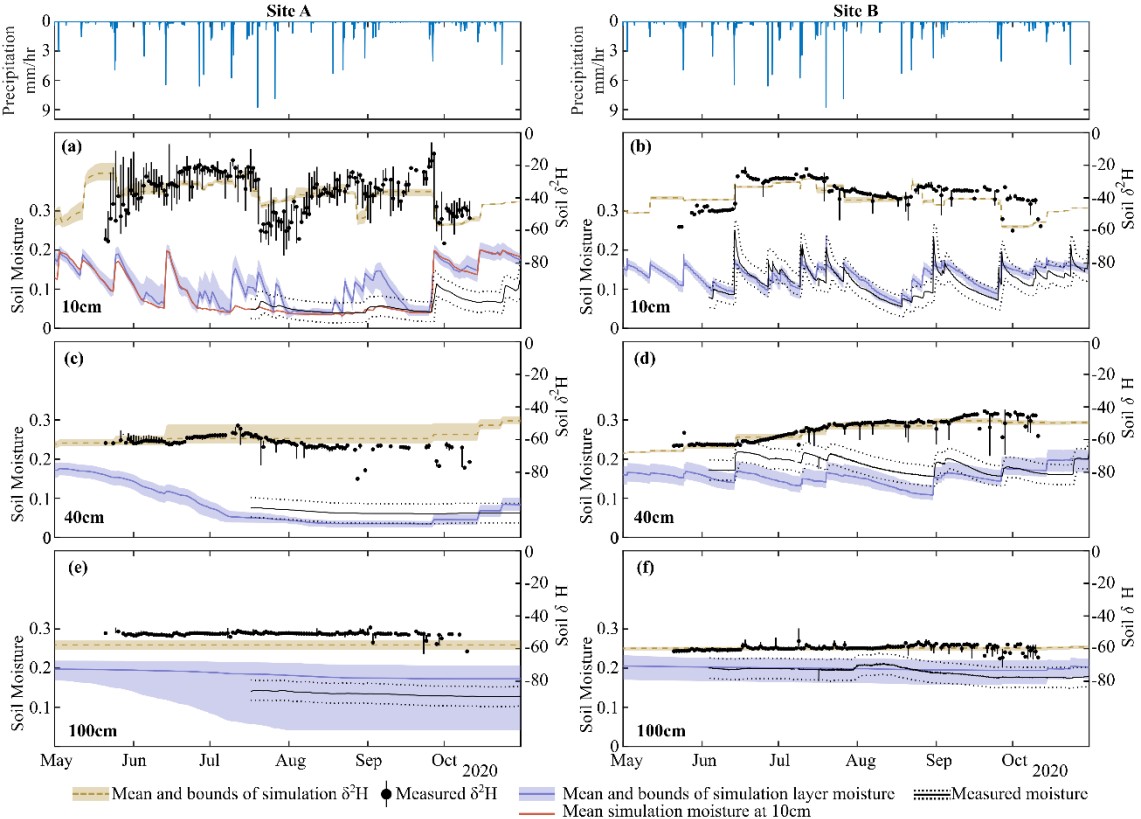

**Figure 3: Simulation mean and upper and lower bounds of soil moisture and δ²H at Site A at (a) 10 cm, (c) 40cm, and (e) 100cm, and at Site B at (b) 10cm, (d) 40cm, and (f) 100cm. The top row shows the hourly precipitation for both sites. Isotopic values (circles) are average daily measurements, with bars showing the daily range in isotopic measurements. δ¹⁸O simulations are shown in Figure S3.**

Estimated energy balance (latent heat, sensible heat, transpiration flux, and temperature) was shown to match *in-situ* measurements within the limits of measurement uncertainty (Fig. 4, Table S2). Despite only utilizing latent heat and surface temperature at Site B in the calibration, simulated sensible heat and soil temperature at multiple depths were reasonably captured (MAE: 4.5W/m², 1.2-2.8°C for sensible heat and soil temperature range respectively, Table S2). Latent heat and sensible heat at Site A (Willow 2) were noticeably more variable than at Site B, driven by the greater transpiration demands of the willow relative to the grasses. Sap flux in the willow was adequately captured by the model (MAE = 0.02m³/day), with the primary deviation - under-estimation - from measurements during the late growing season (Sept-Oct). Greater variability of surface and soil (10cm, Figure S4) temperature was additionally simulated at Site A; however, estimated differences of ground heat storage between Site A and B resulted in increased damping of soil temperature with depth at Site A. Progressive over-estimation of soil temperature during the early simulation period with increasing soil depth is likely due to propagation of a slight over-estimation of early-growing season surface temperature at each site (Fig. 4) potentially due to lower estimated leaf area index at the beginning of the growing season (Fig. 5).

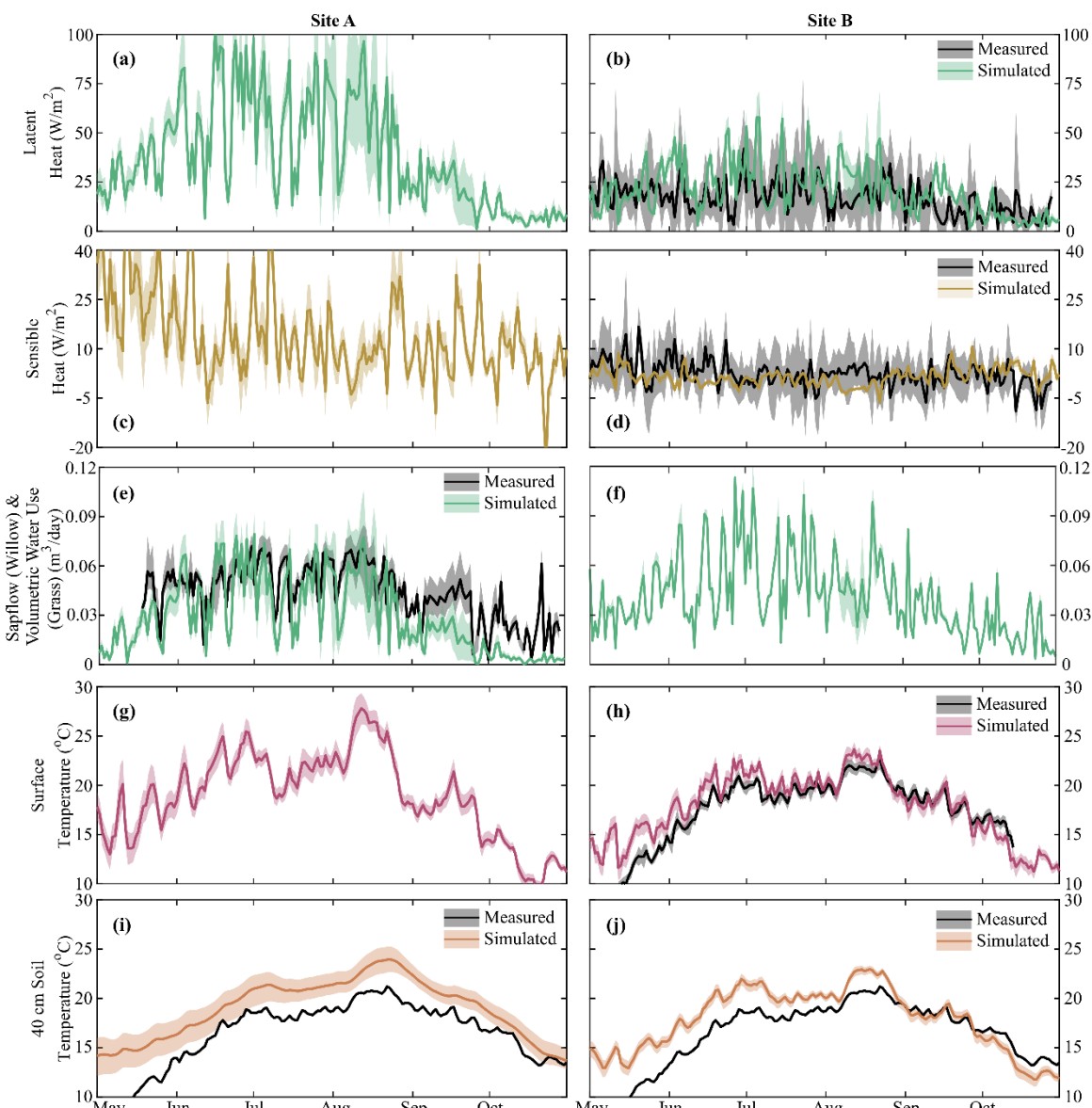

**Figure 4: Simulation mean and upper and lower bounds of energy balance components of (a&b) latent and (c&d) sensible heat, (e&f) volumetric vegetation water use, and (g&h) surface and (i&j) soil temperature (40cm, 10 and 100cm in Fig S4). Black lines show the mean of measured data with grey shaded areas showing uncertainty and sub-hourly variability. Volumetric vegetation water at Site A shows sapflow within the willow, and at Site B shows total volumetric water use of the grass. Note: soil temperature was not calibrated, and transpiration rates are shown in Fig. S4, and sapflow for Willow 1 is identical to Willow 2 (Fig. 4e).**

### 4.2 Growing season dynamics – fluxes and biomass accumulation

Simulated biomass production for foliage and tree diameter produced reasonable results for the two willow trees throughout the simulation period, with only minor deviations (willow LAI MAE = 0.5 m$^2$/m$^2$, Table S2) from both measured and remote sensed (MODIS) datasets (Fig. 5a&b). Average simulations (solid lines) under-estimated LAI at the beginning of the growing

season; however, these differences were relatively small and uncertainty bounds of simulated LAI were within measurement uncertainty. Calibrated grass leaf area index dynamics were consistent with MODIS, with only a small deviation in leaf area index at the beginning of the growing season (MAE = 0.1 $m^2/m^2$; Table S2; Fig. 5c). The dynamics and total net stem growth was adequately captured by the model for each willow (MAE = 1.5-2.2 $\mu m$; Table S2); however, there was a slight underestimation of average net growth in Willow 2 (Fig. 5c&d).

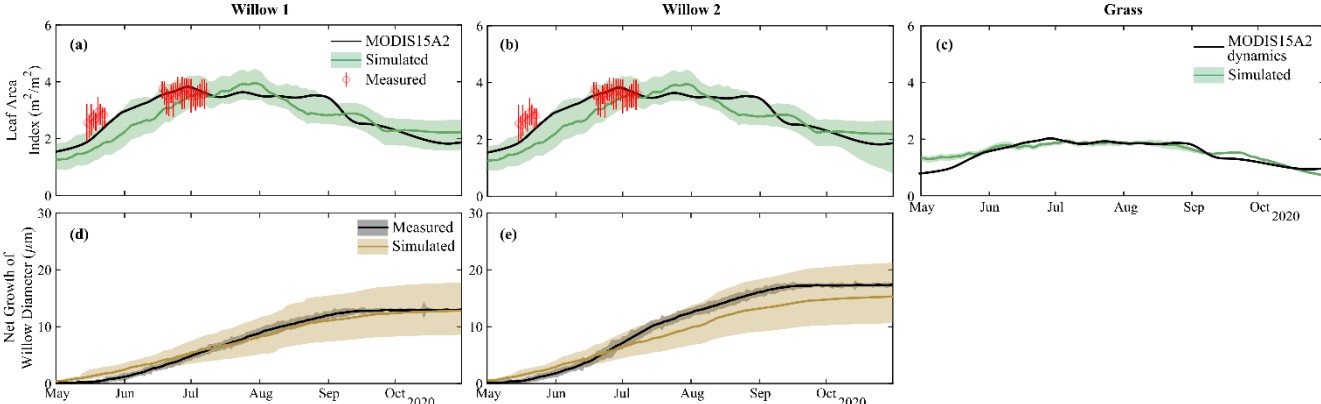

**Figure 5: Simulated and measured leaf area index of (a) Willow 1, (b) Willow 2, and (c) Grass, and net growth of willow diameter of (d) Willow 1, and (e) Willow 2.**

Modelled partitioning of water fluxes and biomass allocation in the willow and grass sites showed marked differences despite the relative proximity (Table 3). Evapotranspiration (ET) dominated water partitioning throughout the summer, with the greatest ET between June 15 and September 15. Total ET was dominated by transpiration ($T_r$) particularly for the willows (>70%, Table 3). This coincided with a decreased proportion of interception evaporation ($E_i$), and soil evaporation ($E_s$) from drier soils. Pre-growing season infiltration was higher below the willows relative to the grass, with moderately lower infiltration below the willows during the growing season. The lower density of grass coverage (and increased bare soil) below the willow reduced growing season differences in infiltration between the willow and grass sites. The high root uptake rate of the willow resulted in negligible percolation of water to deeper soil layers during the growing season. Continuation of drier conditions below the willow resulted in a gradual shift toward near-surface root-uptake compared to a higher proportion (24%) from deeper soils at the beginning of the growing season (Table 3). The willows showed little change in biomass allocation with only a slight shift of root growth in the pre-growing season to foliage growth during the peak growing season (Table 3). Allocation to stem growth showed negligible change throughout the growing season.

The grasses had much higher percolation to deeper soils relative to the willows as a result of low ET. Wetter soils below the grasses resulted in greater soil evaporation proportions. Higher soil moisture throughout the soil profile resulted in more stable root-uptake proportions from layers 1 and 2, with an approximately equal mixture of near-surface and mid-depth (40cm) soil water. Biomass allocation in grasses was static throughout the simulation (as a result of EcH$_2$O's structure), showing a greater allocation of carbon to foliage growth than roots (note there is no allocation to stem growth).

Table 3: Mean (and standard deviation) of water partitioning before (Jan –May), during early growing season (May 1 – June 15), mid growing season (June 16 – Aug 1), and late growing season (Aug 1 – Oct 31). Also shown are the root-uptake (RU) proportions from each soil layer (L1, L2, L3 are layer 1, 2, and 3, respectively) and biomass allocations. Depth of the L1, L2, and L3 are 0-10, 10-40, and 40-100cm, respectively. Note that infiltration and percolation are cell averages, ET is specific to vegetation (proportion of willow in cell < 100%). ET is evapotranspiration, Es is soil evaporation, Tr, is transpiration, and Ei is interception evaporation.

| | Fluxes | | | | | |
|---|---|---|---|---|---|---|
| | Willow | | | Grass | | |
| | Infiltration (mm) | Percolation (L1→ L2) (mm) | ET (mm) | Infiltration (mm) | Percolation (L1→ L2) (mm) | ET (mm) |
| Jan – May | 173.4±2.6 | 127.1±4.9 | 76.9±7.6 | 168.4 ± 1.4 | 128.3 ± 1.0 | 86.5 ± 1.9 |
| May 1 – June 15 | 44.3±1.5 | 4.3±2.9 | 87.8±6.8 | 46.6 ± 0.8 | 16.1 ± 1.4 | 62.0 ± 1.5 |
| June 16 – Aug 1 | 42.9±2.6 | 0±0 | 136±11.3 | 49.8 ± 1.3 | 15.6 ± 1.2 | 87.8 ± 1.7 |
| Aug 1 – Sept 15 | 28.8±1.7 | 0±0 | 95±11.1 | 43.6 ± 0.9 | 21.0 ± 0.7 | 62.0 ± 2.3 |
| Sept 16 – Oct 31 | 41.7±1.4 | 15.2±2 | 35.2±3.9 | 48.2 ± 0.5 | 28.1 ± 0.5 | 34.7 ± 0.8 |
| | Evapotranspiration Fractions | | | | | |
| | $E_s$/ET | $T_r$/ET | $E_i$/ET | $E_s$/ET | $T_r$/ET | $E_i$/ET |
| Jan – May | 0.3±0.05 | 0.33±0.03 | 0.37±0.05 | 0.29 ± 0.01 | 0.34 ± 0.02 | 0.37 ± 0.02 |
| May 1 – June 15 | 0.1±0.02 | 0.7±0.03 | 0.2±0.02 | 0.23 ± 0.01 | 0.54 ± 0.02 | 0.23 ± 0.01 |
| June 16 – Aug 1 | 0.01±0 | 0.77±0.03 | 0.21±0.02 | 0.17 ± 0.01 | 0.52 ± 0.02 | 0.31 ± 0.02 |
| Aug 1 – Sept 15 | 0.01±0 | 0.78±0.03 | 0.21±0.03 | 0.16 ± 0.01 | 0.55 ± 0.02 | 0.29 ± 0.02 |
| Sept 16 – Oct 31 | 0.12±0.03 | 0.4±0.04 | 0.48±0.05 | 0.34 ±0.01 | 0.34 ± 0.02 | 0.32 ± 0.02 |
| | Root Uptake Proportions from Depth | | | | | |
| | RU[0-10cm] | RU[10-40cm] | RU-L3[40-100cm] | RU[0-10cm] | RU[10-40cm] | RU-L3[40-100cm] |
| Jan – May | 0.76±0.02 | 0.23±0.02 | 0.01±0 | 0.46 ± 0.04 | 0.49 ± 0.03 | 0.05 ± 0.02 |
| May 1 – June 15 | 0.76±0.04 | 0.23±0.04 | 0.01±0 | 0.48 ± 0.04 | 0.45 ± 0.02 | 0.06 ± 0.02 |
| June 16 – Aug 1 | 0.83±0.09 | 0.15±0.09 | 0.02±0.01 | 0.52 ± 0.04 | 0.42 ± 0.03 | 0.07 ± 0.02 |
| Aug 1 – Sept 15 | 0.88±0.1 | 0.06±0.08 | 0.05±0.04 | 0.48 ± 0.04 | 0.44 ± 0.02 | 0.08 ± 0.03 |
| Sept 16 – Oct 31 | 0.92±0.06 | 0.04±0.02 | 0.04±0.05 | 0.49 ± 0.04 | 0.45 ± 0.03 | 0.06 ± 0.02 |
| | Biomass Allocation | | | | | |
| | Foliage | Stem | Roots | Foliage | Stem | Roots |
| Jan – May | 0.76±0.17 | 0.05±0.03 | 0.2±0.15 | | NA | |
| May 1 – June 15 | 0.79±0.15 | 0.03±0.02 | 0.18±0.15 | | NA | |
| June 16 – Aug 1 | 0.8±0.14 | 0.03±0.02 | 0.17±0.14 | 0.57 ± 0.31 | NA | 0.43 ± 0.31 |
| Aug 1 – Sept 15 | 0.78±0.15 | 0.03±0.02 | 0.19±0.15 | | NA | |
| Sept 16 – Oct 31 | 0.77±0.16 | 0.03±0.02 | 0.2±0.16 | | NA | |

## 4.3 Evaluation of xylem water mixing

To avoid influencing the evaluation of tree water mixing by the results of simulated soil isotopes (lower variability than measured, Fig. 3a), tree xylem-water mixing was conducted in separate calibrations using both simulated and measured soil water isotopes. Instantaneous mixing of root-uptake water throughout the tree (instant uniformity in all xylem at each time-step, as simulated by the EcH$_2$O-iso model structure) was able to capture the general seasonal dynamics of xylem water in both willows (Fig. 6a&c), though simulated diurnal variability could not fully reproduce the measured variability. Unsurprisingly, the instantaneous mixing approach showed improved performance when measured soil water isotopes were

used rather than simulated soil water isotopes, increasing seasonal and day-to-day variability (Table 4; Fig. 6b&d). The lower AIC for the distance-based mixing simulations of xylem water at 1m suggested that the additional parameterisation aided in the simulation performance of xylem water isotopes relative to the instant mixing simulations. The decrease in KGE in Willow 1 from simulated to measured data (Table 4) was an artefact of the shorter measured soil water isotope time-series (stopping

445    before the end of October) which did not encompass the large depletion at the end of October (time-series length incorporated in AIC). Differences in the AIC and KGE results using measured datasets relative to simulated datasets (Table 4) were due to differences in the method of goodness-of-fit, whereby squared differences (in AIC) were minimized by the distance-based approach but were accompanied by a penalisation of the variance coefficient (in KGE) due to more damped dynamics than the instant mixing approach.

450    Importantly, xylem simulations with instant-mixing and modelled soil isotopic composition were unable to capture the observed standardized hourly variability of the xylem isotopic composition (Fig. 6a&c, right panels). This is consistent with the limited standardized hourly variability of simulated soil water isotopes relative to measured soil water isotopes (Fig. 3a, Fig. 6a&c). Distance-based simulations with modelled soil water isotopes could reasonably capture the timing of sub-daily dynamics due to the sub-daily variability of sap flow volumes. Xylem water mixing estimations using measured soil water

455    isotopes revealed a reasonable standardized hourly variability of xylem isotopes (Fig. 6b&d). Instant mixing unsurprisingly showed dynamics similar to soil water isotopes, which peaked before the standardized measured xylem water. Distance-based mixing with measured soil water isotopes was able to capture the sub-daily variability and timing of the peak; however, the rapid decrease (and late peak) in sub-daily standardized soil water isotopes at 10cm slightly degrades the simulated dynamics late in the day.

460    Model performance improved when the mixing model was calibrated to daily compared to sub-daily (6-hourly or 12-hourly) data (higher KGE). While simulations show deviations from the temporal averaged xylem water isotopes (e.g. 12-hour average, Fig 6), simulations were generally within the minimum and maximum measured range over the time-step. The increase in model performance is consistent with the reduced variability of measured xylem with increasing time-steps, increasing in similarity to the variability observed in soil water isotopes. Likewise, the KGE increased most rapidly with time-step on

465    average when simulated soil water isotopes were used in mixing as they lower variability than measured soil water isotopes (Fig. 3).

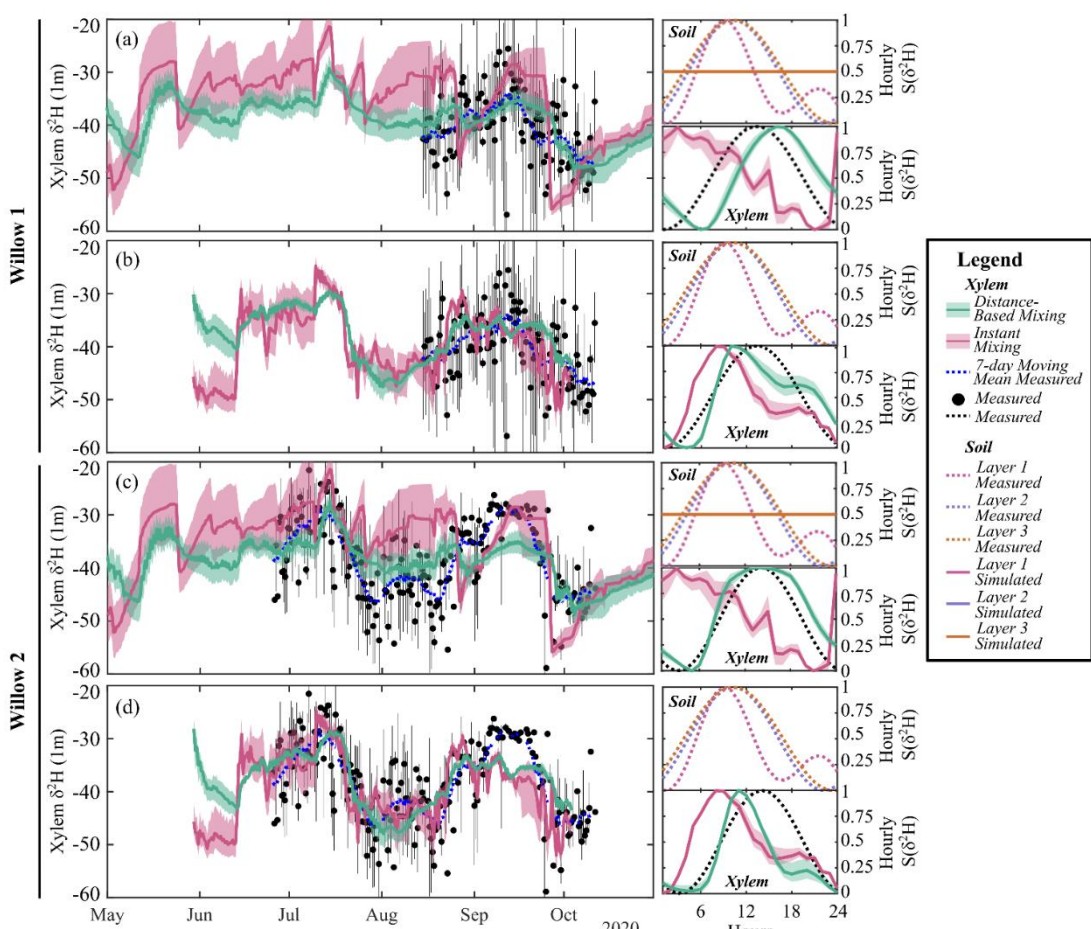

Figure 6: Simulated 12-hour average xylem deuterium using (a) simulated and (b) measured soil deuterium with distance-based and instant mixing in Willow 1. Simulated 12-hour average xylem deuterium using (c) simulated and (d) measured soil deuterium with distance-based and instant mixing in Willow 2. Measurement bars represent the maximum and minimum measured xylem water isotopes during the 12-hour period. For each simulation, standardized sub-daily $\delta^2H$ ($S(\delta^2H)$) shows the measured and simulated soil (cell contribution weighted average) and xylem average hourly variability. Note: standardized simulated soil water isotopes overlap for all soil layers.

Table 4: Akaike information criteria (AIC) and average KGE (±standard deviation) for model performance for each willow evaluated at different time-steps. Values in parentheses for KGE show the KGE of sub-daily variability. Mixing models used either output (isotope and sap flow) from EcH₂O-iso or measured isotope and sap flow. All models used estimated uptake proportions from EcH₂O-iso calibrations.

| | Time Step | | 6hour | | 12hour | | 24hour | |
|---|---|---|---|---|---|---|---|---|
| | Willow | | 1 | 2 | 1 | 2 | 1 | 2 |
| **EcH2O-iso modelled isotope** | **Distance-Based** | **AIC** | 1577.7±17.1 | 2104.0±32.9 | 800.3±8.9 | 1209.2±21.3 | 367.2±6.3 | 551.2±31.3 |
| | | **KGE** | 0.11±0.02 (0.80±0.03) | 0.27±0.02 (0.83±0.03) | 0.31±0.02 (0.48±0.09) | 0.34±0.02 (0.80±0.02) | 0.36±0.02 (0.73±0.04) | 0.42±0.02 (0.84±0.03) |

| | | | | | | | | |
|---|---|---|---|---|---|---|---|---|
| and sap flow | Instant | AIC | 1822.3±40.3 | 2566.8±93.3 | 951.9±25.4 | 1584.3±62.8 | 519.8±15.8 | 900.1±40.0 |
| | | KGE | 0.15±0.04 (-0.12±0.05) | 0.30±0.04 (-0.16±0.05) | 0.12±0.09 (-0.12±0.05) | 0.35±0.06 (-0.16±0.05) | 0±0.15 (-0.12±0.06) | 0.41±0.09 (-0.16±0.05) |
| Measured isotope and sap flow | Distance-Based | AIC | 1403.5±17.0 | 2044.3±24.8 | 656.0±16.7 | 1159.1±22.2 | 314.5±9.5 | 588.7±21.5 |
| | | KGE | 0±0 (0.51±0.04) | 0.39±0.04 (0.53±0.04) | 0.14±0.0 (0.55±0.05) | 0.53±0.04 (0.56±0.04) | 0.36±0.01 (0.61±0.02) | 0.60±0.05 (0.54±0.05) |
| | Instant | AIC | 1415.4±22.8 | 2099.2±27.9 | 683.3±19.0 | 1236.2±22.7 | 334.4±15.9 | 645.1±19.1 |
| | | KGE | 0.22±0.02 (0.59±0.06) | 0.47±0.02 (0.51±0.07) | 0.33±0.02 (0.59±0.06) | 0.54±0.03 (0.51±0.07) | 0.45±0.04 (0.59±0.06) | 0.63±0.03 (0.51±0.07) |

## 4.4 Evaluation of fluxes, uptake, and water ages through the growing season

Differences in infiltration and percolation quantities between the willow and grass (Table 3) further propagated to water ages below each vegetation type (Fig. 7a&b). Average layer 1 water ages below the willow (15 ± 1 days) were similar to the grass (17 ±1 days), with larger proportions of 7-day (36±2%), 14-day (59±2%), and 30-day (84±1%) relative to those below grass (31±1, 52±1, and 67±1%, respectively). The higher proportions of young water (more recent precipitation) below the willow were primarily the result of lower moisture storage, causing a greater effect of infiltration on water ages. The limited percolation to layer 2 below the willow resulted in a continuously increasing water age in the deeper soil water. Percolation throughout the summer under the grasses resulted in a stabilization of water age (average: 73 ± 2 days), with 81±1% of water older than 30 days. Average root water age in the root-tip was a mixture of water ages for all soil layers, with the high root uptake from shallow soils willow resulting in a younger average uptake water age in the willow roots (28 ± 6 days) compared to the more equal mixture of shallow and mid-depth soils uptaken by the grass roots resulting in older water uptake (51 ± 3 days) (Table 3).

Water ages in storage and uptake were further discretized into contributions of water from each monthly precipitation amount to characterize the seasonal origins of water stored and utilized. Fast turnover of water in the shallow soils (i.e. upper 10cm) resulted in low percentages of late winter and spring water (Pre- April 1) in the early growing season (13 and 18% below willow and grass, respectively, Fig. 7, Table S3&S4), which were negligible at the end of the growing season (0 % below willow and grass, Table S3&S4). Limited percolation from shallow to deeper soils under the willow resulted in large proportions of deeper water originating from spring or winter precipitation (77%), while under grass the spring and winter precipitation only accounted for 37% of deeper soil water by the end of the growing season. Differences in rooting distribution and the temporal dynamics of infiltration resulted in differences of spring and winter water usage for the grass and willow throughout the growing season, with 33, 16, 9% for early, mid-, and late growing season willow (57, 35, 24%, respectively for grass) root-uptake of spring/winter water.

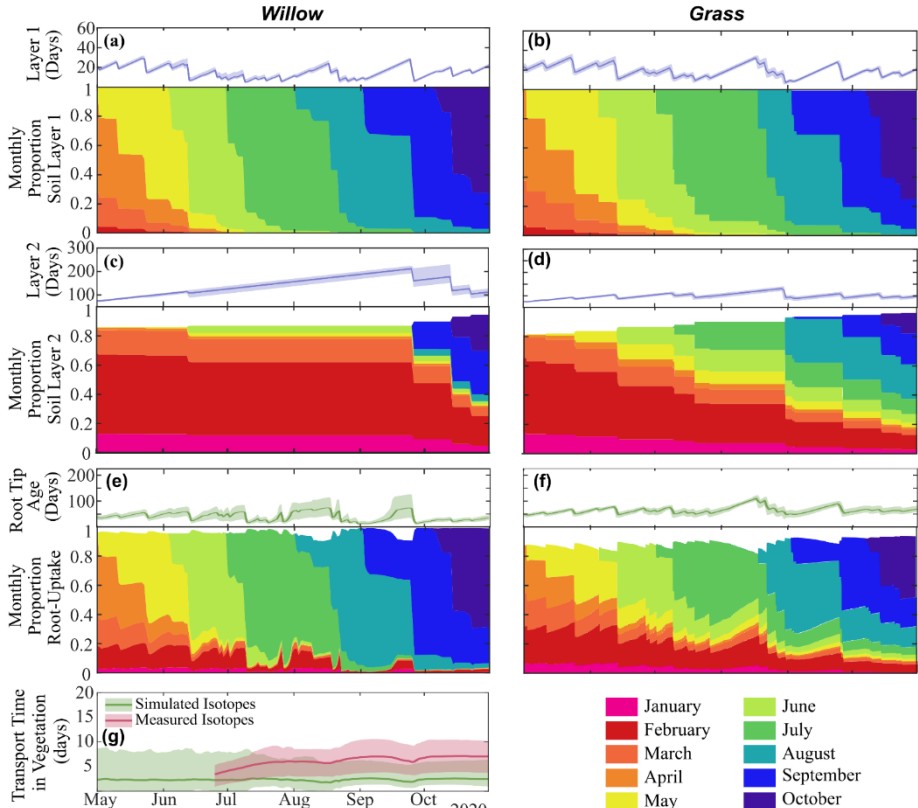

**Figure 7: Average water age (and range) and fractional monthly water contribution in layer 1 (10cm) in (a) willow 1 and (b) grass, in layer 2 (40cm) in (c) willow 1 and (d) grass, average (and range) of water entering roots in (e) willow 1, and (f) grass, and (g) the average (and range) of time between root uptake and 1m using distance-based mixing using simulated or measured soil water isotopes. Fractional water ages shown in Fig. S5 and transit times from different soil layers in Table S5.**

Since instantaneous mixing equates water age through all vegetation storage and pathways (i.e. roots and xylem), xylem ages are only shown for the distance-based mixing approach (using simulated and measured soil water isotopes). Using simulated soil water isotopes, distance-based mixing revealed an average transit time (all time-steps) of $149 \pm 30$ hours from the roots to 1m height (Fig. 7g). Distance-based xylem mixing using measured soil water isotopes revealed a mean transit time of $237 \pm 97$ hours (Fig. 7g). Smaller uncertainty of transit time estimated from simulated soil and sap flow was a direct result of smaller temporal variability (lower degrees of freedom) of simulated soil water isotopes compared to measured soil water isotopes.

## 5 Discussion

### 5.1 Assessing growing season flux-storage-tracer relationships, water partitioning and biomass dynamics

The utility of ecohydrological modelling as a tool to evaluate the linkages between water partitioning and biological dynamics is directly related to the capability of such models to accurately reproduce a wide range of variables to indicate acceptable

representation of ecophysiological processes. At fine spatial scales, such as plot sites, reproducing flux-storage-biomass dynamics and feedback mechanisms are important for the transference of their nonlinearities to larger scales (Asbjornsen et al., 2011). Advancements of complex (e.g. physically-based) models and fusion with multiple, high temporal resolution data streams can lead to the convergence of model performance for all variables (Asadzadeh et al., 2014) and higher confidence in the ability of the model to perform across a wide range of hydroclimatic conditions. The application of a tracer-aided ecohydrological model on sub-daily time-steps using multiple data streams from an intensive *in-situ* monitoring site provided a means to directly evaluate how well the model could simultaneously reproduce flux-storage-tracer-biomass dynamics through multiple criteria throughout the high variability of a growing season.

Overall, the model performed well against all measured data in both the willow and neighbouring grass sites, including soil water storage, water and energy fluxes, isotope variations, and biomass dynamics throughout the growing season. Total green water usage (total ET) was notably higher for the willows compared to the grass, similar to other nearby studies (Douinot et al., 2019; Kleine et al., 2021; Smith et al., 2020b). Further, the proportions of ET for interception evaporation (as a proxy for total interception) and transpiration were well within expected ranges for both vegetation units (0.4-0.8 = Tr/ET) (Coenders-Gerrits et al., 2014; Dubbert et al., 2014; Schlesinger and Jasechko, 2014; Zhou et al., 2016), including lower Tr/ET for the grass compared to the willows. This suggests adequate reproduction of water partitioning within the model. Despite the relatively dry characteristics of the soil at and below the soil moisture measurement at 10cm, transpiration in both willows and grass were maintained throughout the study period (Fig. 4), albeit with an under-estimation of the transpiration in the willows toward the end of the growing season. The under-estimation was accompanied by an over-estimation of soil moisture but appropriate depletion in the shallow soils below the willow (Fig. 3a). While the late season deviation of estimation of sapflow and soil moisture are directly linked, the model structure does not permit temporal variability in lateral root water usage (i.e. outside the model cell), therefore increased late-season transpiration could have detrimental effects on shallow soil moisture below the grass. Furthermore, an increase in late-season transpiration would only be the result of vegetation parametric changes outside feasible ranges. The high goodness-of-fit for the primary growing season for the willow trees was mainly possible within EcH$_2$O-iso due to the capability of the willows to use water from adjacent model cells. As observed in other nearby studies, soil evaporation contribution to ET under both the willow and grass was damped throughout the growing season due to leaf shading (LAI) reducing energy availability at the surface (Kleine et al., 2021; Smith et al., 2021). Shading effects of the willows with higher interception capacities could be the cause of the late-season stability in isotopic measurements below the grass due to reduced precipitation input compared to the simulated large depletion of measured and simulated soil water isotopes below the willows. Although willows are generally a riparian species able to access groundwater or surface water, the willows at this site did not reveal contributions from the groundwater (deeper than *in-situ* soil water isotope measurements, >2m). The predominantly shallow root-uptake contribution (driven by higher near-surface water availability and root-distribution) was consistent with root distributions observed in willows elsewhere (Cunniff et al., 2015; Phillips et al., 2014), and only slightly lower than independent Bayesian mixing estimations conducted using soil and xylem isotopes for the same trees (Landgraf et al., 2021). The near-equal contribution of shallow and mid-depth uptake in the grass (and subsequent older

water age in mid-depth soils) was more prominent than expected given the shallow rooting zone. The deeper rooting distribution in the grass is likely due to a combination of vegetation functional types (i.e. willow v. grass) and high uptake competition in the near-surface soils with the nearby high water-use willows (Moustakas et al., 2013). Competition could potentially drive deeper rooting of the grass, contributing to the more dynamic soil moisture at 40cm as measured below the grass. The slight decrease in simulated root biomass production in the willows during the summer months suggested that vegetation was not under water stress in the study period as allocation increased for leaf (water usage) rather than root (water sourcing) (Eziz et al., 2017). Average hourly simulated transpiration dynamics were comparable to sap flow measurements (Fig. S6), including small, but notable uptake during the night, similar to observations in other woody plants (Dawson et al., 2007).

Within this study, the relatively low soil moisture did not propagate to a notable decrease in sap flow. While the low shallow soil moisture and the high proportion of near-surface roots of the willows seemed inconsistent with the measured high vegetation water usage, *in-situ* isotope dynamics were essential in confirming measurements of shallow moisture were not underrepresenting infiltration events (i.e. via spatial heterogeneity). Multicriteria calibration using high-temporal resolution and highly dynamic soil water isotopes revealed a strong positive correlation of simulated shallow soil moisture KGE and simulated shallow soil water isotope KGE (i.e. model performance of soil water isotopes only improved with the performance of soil moisture). Bulk soil water isotope samples, with a coarser temporal resolution, would have been unlikely to adequately capture the necessary isotope dynamics for calibration. EcH$_2$O-iso was only able to adequately estimate soil moisture at 10cm through sub-discretization of the soils above 10cm, which suggested high retention of water in soils above the measurement depth (e.g. 0-5cm), up-taken by roots or soil evaporation before there is sufficient water to percolate to 10cm. These small-scale spatial variations, as well as the large spatial differences between the soils below the willows and below the grass, and between different soil layers (Fig. 3) reveal the significant heterogeneity of the site despite relatively immature soils and the local spatial scales. The heterogeneity of moisture and isotopes is likely further exacerbated by spatial heterogeneity under the canopy, as differing branch and leaf structures can impact detailed throughfall distribution (Dalsgaard, 2007; Gerrits et al., 2010; Li and Liang, 2019). If such data were available, it could likely further improve soil moisture simulations as zones of higher soil moisture may be present at locations below the canopy where throughfall is concentrated (Gerrits et al., 2010) which could in turn influence and increase dynamics of near-surface soil water isotopic compositions. These heterogeneities would likely help improve the model performance of near-surface soils by accounting for additional variability in measurements. However, given the relatively small quantity of water encompassed in the shallow soils relative to total fluxes, further heterogeneity of measurements in soils would not likely influence the overall estimation of evapotranspiration fluxes (or partitions), or root water sources.

## 5.2 Evaluation of mixing dynamics of root-uptake and implications of the rooting zone

We used output from EcH$_2$O-iso (i.e. the root-uptake sources) and a simple, parsimonious approach to further explain the time-delay and variability of xylem water isotopic composition from soil water isotopic composition. The approach showed

substantial improvements over the use of the instantaneous mixing currently applied within most modelling approaches. While the instantaneous mixing had occurrences of higher KGE than the distance-based approach, this was the product of greater simulated variability rather than improved dynamics, as indicated by the AIC. Further, the additional capabilities of the distance-based model to capture the lagged response of diurnal soil water isotopes in xylem further underlines the advantages of the approach. The inclusion and calibration of rooting length into EcH$_2$O-iso showed comparable results to other willows (1.3-1.4 v. 1.3 tree height/root length; Table S5; Phillips et al. (2014)) which supports the initial estimation of spatial uptake sources. With around half of the uptake (by root length and water availability) estimated to occur outside of the willow cells, this consideration of spatial variability in soil water isotope composition underlines the potential influence of spatio-temporal variability of source waters on xylem isotopes, as suggested by De Deurwaerder et al. (2020) and Seeger and Weiler (2021).

As with partitioning soil moisture and vegetation water usage, the high-resolution *in-situ* isotopic composition of xylem water was indispensable for constraining the mixing dynamics of root-uptake. Coarser temporal destructive xylem sampling could potentially over-estimate xylem variability due to the large sub-daily variability, damping seasonal patterns (e.g. enrichment in September, Fig. 6) and influencing estimated root-uptake profiles. Further, while the use of tracer-aided ecohydrological model output for estimating xylem water composition with refined rooting distributions was able to reconcile general seasonal xylem dynamics and some sub-daily variability due to sap flow (7-day moving average and sub-daily variability, Fig. 6), deviations in seasonal magnitudes of simulated to measured xylem water isotope dynamics were primarily due to differences in simulated v. measured soil water isotopes (e.g. Fig. 6a v. Fig. 6b) in the shallow soils (Fig. 3 & RU in Table 3). This additionally highlights the importance of *in-situ* isotopic soil and vegetation measurements to constrain model estimates. The use of measured data within the same framework further showed the capabilities of the approach to estimate xylem dynamics, including more direct translation of soil sub-daily variability in the xylem (Fig. 6) as suggested by von Freyberg et al. (2020). However, even the use of *in-situ* measured soil water isotopes was unable to fully resolve the xylem dynamics with much larger day-to-day variability than could be estimated by the modelling approach (Fig. 6, Table 4).

Uncertainty in isotopic measurements, complex physiological processes within the trees which may cause fractionation, limitations of the model structure and interactions between all of these were potentially limiting factors for reducing the xylem mixing model performance. *In-situ* measurement uncertainty, particularly in xylem, may be large for $\delta^2$H (5-10 ‰; Beyer et al. (2020)), greater than the standard deviation of the daily mean $\delta^2$H in xylem (3.7 ‰). Variation in the daily mean is additionally lower than the sub-daily standard deviation (6.8 ‰) which would suggest sub-daily variations are more significant than daily. These daily variations in xylem $\delta^2$H could be influenced by natural short-term climatic variability and monitoring errors, with condensation, mixing and diffusion potentially causing instabilities in measurements and greater uncertainties (Beyer et al., 2020; von Freyberg et al., 2020). In this study, condensation uncertainties were minimized by heating the tubing and were largely restricted to the autumn (September and October) when nighttime temperatures could be much cooler (Landgraf et al., 2021). Further, while fractionation during uptake or within vegetation has frequently been discussed and re-examined recently (Brinkmann et al., 2019; Poca et al., 2019; Vargas et al., 2017), the vegetation isotopes reflect the fractionated soil water isotope sources (Landgraf et al. (2021)) which suggests that vegetation specific fractionation is not

likely a major cause of isotopic variation on either hourly or daily time-steps at our site. Finally, the model structure may play a role in the xylem mixing estimation, in particular with the assumed "static" transit time (consistent with the SPAC supply-demand models e.g. Sperry et al., 2016a and Simeone et al., 2019) limiting effects of dynamic tree cell water storage exchange with xylem. These water pools may provide a different source of contrasting isotopic composition to sap flow when transpiration is lower (De Deurwaerder et al., 2020; Secchi et al., 2017) and may make a significant contribution to transpiration fluxes at certain times (Urban et al., 2014). The linkage of the model structure of flow path mixing as shown here, and tree cell water storage, inclusive of simple parametric storage mixing approaches (eg. Knighton et al., 2020) presents potential further model development to better estimate sub-daily and inter-daily xylem variability and interpret water use dynamics in vegetation. While such a release of water may be present during peak transpiration hours, as suggested by the hourly variability of measured basal diameter (Fig. S7), the high water usage of willows potentially limits the proportional contribution of cell storage to xylem water.

## 5.3 Implications of water ages in soil water and xylem fluxes

Quantifying the age of water has significant implications for improved understanding of how water is cycled through the critical zone, particularly for understanding the temporal changes and response times of water quantity and quality in different environments (Sprenger et al., 2019). Further discretization of average water ages into age distributions helps to illuminate how the evolution of water ages are influenced by different processes, (Rodriguez et al., 2020) as well as the influence of water fluxes from different temporal periods or extremes (Allen et al., 2019). The modelled contributions of different water ages in the willow root-uptake suggested that summer precipitation was the dominant source of water, accounting for 56% of total uptake (volume-weighted). This differs from isotope-based studies in other locations (e.g. Allen et al. (2019) across Switzerland) which have shown a higher usage of winter precipitation in leafy vegetation (beech and oak), but it is consistent with the estimated high contribution of recent precipitation during the growing season in the study region (Miguez-Macho and Fan, 2021). However, differences in hydroclimate, as well as variations in rooting depths between species and maturity of forests stands, likely give deeper rooting depths for beech and oak compared to shallow roots for the younger willows (in this study) which could drive differences in seasonal precipitation use by vegetation. Alternatively, the effect of separation of storage with fast and slow turnover times, or the effects of preferential flow (Sprenger and Allen, 2020), could contribute to an increase in water age if younger summer water preferentially moves through the soil, or if tightly bound water is used by vegetation. However, the good temporal agreements of soil and xylem isotope samples relative to the likely dichotomy between faster and slower moving isotopes in the soil (e.g. Sprenger et al. (2019)) implies that for the conditions present at the study site, inclusion of preferential flow or variable pore size would not influence water ages for the uptake from either of the soils. This is also consistent with the immature nature of the soils and their relatively uniform sandy nature.

Age estimates for water being transported in xylem derived in this study were of a similar order of magnitude to estimates for other woody vegetation; averaging between 6.2 and 8.1 days with a maximum of 10.3 days at our site compared to between two days and one month in other studies (Brandes et al., 2007; Meinzer et al., 2006). While our estimates were towards lower

transit times compared to most other studies, the willows at our site were younger, had smaller diameters, were measured at 1m basal height compared to in the canopy, and had higher xylem sap flow velocity, consistent with high water usage and low stomatal control (xylem cavitation control, Wikberg and Ögren (2007)). Additionally, differences in xylem water ages relative to previous root-uptake mixing studies (e.g. Seeger and Weiler, 2021) are likely due to stand size, vegetation age, and

measurement heights. High-resolution *in-situ* measurements, capturing soil and xylem isotope dynamics were indispensable for improving the confidence in the estimated xylem transport age. Coarser xylem isotope samples would likely be insufficient to adequately constrain transport age uncertainty. Similar to the impact that stored tree water release can have on xylem isotopes (Section 5.2) (Brandes et al., 2007; De Deurwaerder et al., 2020); there are likely implications for water ages and identification of sources due to water released from cell storage. The effect of stored water on xylem transport ages could be

significant in vegetation that utilizes large fractions of previously stored water (e.g. which can be as much as 20% of transpiration; Čermák et al. (2007)). In such vegetation, total transpiration could be maintained from stored water for a week (Čermák et al., 2007), drawing water from storage cells which may be notably older than xylem water. Effects of such long-term storage components have previously been implicitly incorporated into post-root uptake lumped vegetation mixing approaches (e.g. Knighton et al., 2020) which have previously shown order of magnitude changes in tree stored water age with

changes in wetness conditions. Due to the relatively young uptake water ages of willows in this study, the inclusion of any cell water storage (~16L/day if 20% of transpiration, ~600L total storage to maintain transpiration for one week) may significantly increase the mean age of xylem water and would increase the total vegetation mixing time. Given the limited studies describing the ages of water and tracers in transpiring trees (Sprenger et al., 2019; Mennekes et al. 2021; Benettin et al. 2021), more studies quantifying the transport through the xylem, as conducted here, and the inclusion of additional mixing within vegetation

(e.g. Steppe et al. (2006)) would be beneficial to further constrain plant water use estimations.

**6 Conclusion**

In order to increase understanding of water cycling mechanisms in the critical zone, it is essential to evaluate how water is partitioned by vegetation and the dominant processes controlling water usage and movement. Furthermore, quantifying flux and storage dynamics, through methods such as ecohydrological modelling, are necessary to understand the linkages of water

in vegetation and soil at high spatial and temporal resolution. We used a large *in-situ* dataset under grass and willow trees, including soil moisture, energy balance, water stable isotopes, and biomass accumulation to test the capability of using a tracer-aided ecohydrological model at high temporal resolution to constrain water, energy, and isotope mass balance throughout a growing season. The model captured event and seasonal dynamics of soil moisture (MAE = $0.02-0.03\text{m}^3/\text{m}^3$), soil and xylem water deuterium (MAE = 4.2-8.6‰), latent and sensible heat (MAE = 1.8 – 20 W/m$^2$), and biomass (MAE of stem growth and

leaf growth, 1.5-2.2μm and 0.5m$^2$/m$^2$, respectively). *In-situ* soil and vegetation isotopes were indispensable in calibration for simulating water storage, sources and fluxes of vegetation uptake, mixing processes and water age estimates; and the importance of such datasets in informing modelling approaches was demonstrated. Modelling sapflow of the willows revealed

significant water usage from neighbouring cells, and heterogeneity of root water and xylem water isotope sources. Distance-based isotopic mixing of root-uptake revealed an improved estimation of xylem water stable isotopes and showed the capability
of the model to reproduce the diurnal isotopic variability in spatially heterogeneous soils as a lag in the xylem water isotopes. The results additionally pointed towards further model development needs for modelling vegetation mixing in natural environments. Such numerical modelling approaches, with a physical basis and the capability of accurate simulation of multiple, inter-related variables estimation, have a high potential for further exploration of critical zone water cycling and improved understanding of spatio-temporal changes in water availability due to vegetation-soil interactions. Continuation of
integrated modelling approaches using the leverage provided by *in-situ* data will aid future ecohydrological investigations in constraining and informing modelling while providing high spatio-temporal resolution insights into ecohydrological processes.

## Acknowledgements

The *in-situ* data could be measured through equipment funded by funding by the BMBF (No: 033W034A). Funding for CS was through the project "Modelling surface and groundwater with isotopes in urban catchments" (MOSAIC) provided by the
Einstein Foundation. Contributions from CS were also funded by the Leverhulme Trust's ISOLAND project. We acknowledge the IGB and Leibniz Association Open Access Publication Funds. The authors acknowledge the assistance of David Dubbert, Lukas Kleine, and Jonas Freymüller in isotope analysis and study site set-up, and Marco Maneta for discussions on EcH2O-iso modelling.

## Code availability

The model code of EcH$_2$O-iso is available on Bitbucket at http://bitbucket.igb-berlin.de:7990/users/ech2o/repos/ech2o_iso/browse

## Data availability

The data used in this study are available in the open access data base, FRED (https://fred.igb-berlin.de/data/package/582). Isotope data are password protected, with full access available from the corresponding author upon request.

## Author contributions

AS conducted model set-up, calibration and validation of the EcH$_2$O-iso model mixing model utilizing earlier work led by DT, CS, MD. JL collected and conducted correction, quality control and assessment of the *in-situ* data. AS prepared the manuscript with contributions and editing from all co-authors.

## Competing interests

The authors declare that they have no competing interests.

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
