# Peer review of "Modelling temporal variability of *in-situ* soil water and vegetation isotopes reveals ecohydrological couplings in a riparian willow plot"

_Biogeosciences, 2021_

## Author Response (AR1)

**General Comments: Associate Editor**

Dear authors,

after carefully reading your manuscript, the two reviewers' assessments and your responses, I have come to the conclusion that your manuscript can be considered for publication in Biogeosciences after major revisions.

Both reviewers highlight the novelty of your study, but have identified a number of more or less serious issues that need to be addressed. You have already partially started to address these questions or points in your answer, but they need to be further elaborated.

When reading your manuscript, I was not so sure about the novelty of your study, since there is a very similar study by Seeger & Weiler published in Biogeosciences in 2021, which you cite though (although still the BGD version), but which you do not really compare your approach and results to. Thus, it remains unclear what is really novel (and maybe better solved) in your study. Then I was puzzled about your parameterization and calibration approach, since you obviously deviated from the common split calibration approach (i.e., splitting the dataset in a calibration and validation part) for unclear reasons. You just mention in L305 that "the evaluation of a single growing season at the site limits the feasibility of a split calibration approach", but you do not provide a more detailed reason, and more importantly, no clear alternative, but you simply state that "all available data (except $\delta 18O$) were used to constrain model parameterization." Also, the approach of permitting water uptake from neighboring (but not well-defined) grid cells appears somewhat like an "escape" or "gap-filling" strategy. Where did you get the necessary information from for the adjacent cells?

I also fully agree with Reviewer 2 that the "lack of replication, uncertainty of soil and plant water isotope measurements, and spatial variability of ecohydrological measurements makes the quantitative value of the modeled data at least questionable" and that "an honest evaluation and interpretation of the modeled data in that regard would benefit the manuscript". I also endorse Reviewer 2's recommendation to provide also the $\delta 18O$ data and use the dual isotope space for more precise evaluation of your model data. I also agree that you frequently use subjective statements to describe your results. These statements should be replaced with more quantitative information.

Overall, in view of the indispensability of measured moisture and isotope data for your modeling approach, I was not fully convinced at the end that the use of the ECH2O model comes along with a real advantage over using measured soil profile moisture and isotope as well as xylem isotope data together with simple mixing models and Bayesian root water uptake modeling. This should be made much clearer in the manuscript and in the Conclusion section.

Specific comments:

Your "Response to R1C22": Here you state that "the analytical precision of 2H is better than 18O". Neither for measurements with a Picarro, nor for measurements with an IRMS this statement is true, but the opposite.

Your "Response to R2C9: The assumption of root distribution here is that the root distributions follow an exponential distribution…": But even exponential distributions can have a very different profile, if the parameters of the (same) exponential function are very different.

More specific comments and technical corrections can be found in the annotated manuscript.

**Response to General Comments: Associate Editor**
The authors thank the Associate Editor for their consideration of the reviewer's comments and for their further comments and technical corrections on the pdf.

To address your general concerns raised above:

Novelty in the context of the Seeger and Weiler (2021) study: although this study involves similar use of in situ data and assesses root water uptake, it has an entirely different context and therefore there is not a basis for direct comparison. This is because (a) it involves an experimental manipulation with artificial irrigation and (b) does not use a multi-criteria calibration of a full physically-based ecohydrological model to integrate the in situ data. That said, we do now compare our results for ages of xylem water in the discussion.

Model calibration: as we only have data over one growing season, a traditional formal split calibration/validation approach is not possible. However, this tradition is more from hydrological modelling with single calibration targets (e.g. stream flow, soil moisture etc.). With multi-variate calibration, the skill of the more to simulate multiple times series of water, energy and biomass related data is an exacting test of model performance. In addition, we also use other data streams (e.g. 2 locations and 3 depths at each location for $\delta^{18}O$ and soil temperature, and sensible heat above the grass) for informal validation. It is important to also note that *in-situ* isotope monitoring is highly demanding resource wise and not currently operationalized for multi-year continuous studies.

Root water uptake across cells: allowing this wasn't "closing a gap" but rather using the model-data fusion as a learning framework to identify where the model needs improvement. As noted by Reviewer 1, this development is an advance that has wider implications for future model applications (see below).

Replication and uncertainty: Of course with *in-situ* high-resolution isotope monitoring, replication is limited due to the high capital costs and taxing logistical demands. However, by using two soil pits and two trees, which captured a high degree of heterogeneity and uncertainty

(which were openly shown in the paper), it seems a little unfair to imply the study lacked any replication or consideration of uncertainty. Nevertheless, we have underlined the uncertainty in the data and modelling results in revision. And we have included uncalibrated simulations of 18O in the Supplementary Material for the dual-isotope assessment.

Need for process-based modelling: although, in this particular case, the EcH$_2$O-iso application yielded similar root-water source apportionment to a simple mixing model (reported in Landgraf et al., 2021), the range on insights is immeasurably richer because of the integration of energy fluxes, water budgets and biomass accumulation. For example, estimation of water age distributions isn't possible from a simple mixing model.

To address these and other issues in the revision, the authors have clarified the significance of the study (objectives), differentiation from previous studies (methods), model development and calibration (methods), and the interpretation of results (including measurement uncertainties), and have improved the discussion by adding the further description of current limitations and potential paths forward for ecohydrological modelling. In particular, the inclusion of measurement uncertainties have increased the capabilities of a quantitative assessment of the model performance and demonstrated the replication of the plant and soil water isotopic measurements. The revisions have further clarified the importance of using ecohydrological models throughout the manuscript, and have highlighted the motivation and advantages of utilizing physically-based model approaches for this study.

With the extensive revisions that have been made to this manuscript by the three very detailed reports, the authors hope that the revised manuscript is acceptable for publication in Biogeoscience.

**Specific Comments: Associate Editor**
**AEC1**: water
**Response to AEC1**: Revised L150

**AEC2**: Please provide product specifics, such as wall thickness and pore size, and manufacturer information.

**Response to AEC2**: For the sake of brevity, the authors have included some of the relevant information and have provided a reference to Kübert et al. 2020 for further details of the polypropylene membranes. However, we also refer to the companion paper by Landgraf et al. (2021) HESS-D paper where the details of the set-up are given. L151

**AEC3**: isotope
**Response to AEC3**: Revised L150

**AEC4**: See comment above
**Response to AEC4**: See **Response to AEC2**

**AEC5:** with
**Response to AEC5**: Revised L161

120

**AEC6:** for
**Response to AEC6**: Revised L161

**AEC7:** Provide dimensions \(depth in the stem, diameter\).

125   **Response to AEC7**: The authors have provided the diameter of the borehole, and for the sake of brevity have added the reference to the borehole method to further descriptions of the approach used in this study site. L160

**AEC8**: With which material?

130   **Response to AEC8:** Commercial silicon was used to create the seal. This has been revised. L163

**AEC9:** How did you decide whether there was a wounding effect?
**Response to AEC9:**Wounding effects were identified by daily visual inspection of the trees and clear changes in the isotopic compositions which were unrealistically high and revealed an

135   exponential decrease in time to more stable isotopic compositions. L164-165.

**AEC10:** extractions
**Response to AEC10:** Revised L166

140   **AEC11:** Flushed with what? Ambient air? Dry air? Synthetic air?
**Response to AEC11:** The line was flushed with dry air each morning. The authors have revised this in the manuscript. L167
**AEC12:** Which regression function was used?
**Response to AE12:** A nonlinear polynomial regression was utilized for regression. This has

145   been revised in the manuscript. L169

**AEC13:** Here and throughout the ms: "soil isotopes" is a sloppy term. There are many isotopes of many elements in soil \(hydrogen, carbon, nitrogen, oxygen, sulfur, calcium, iron, uranium etc.\). Be specific: "soil water isotopologues" or "soil water isotopic composition" or the like.

150   **Response to AEC13:** OK, though it is implicit that the study exclusively considers water isotopes. Throughout the manuscript the authors have revised the term "soil isotope" to "soil water isotope".

**AEC14:** KGE,

155   **Response to AEC14:** Revised. L290

**AEC15:** Can you give an estimate from how far beyond the cell the roots could take up water?
**Response to AEC15:** During revision, the authors have provided the maximum extent allowable by the model in Section 3.4.1.

160

**AEC16:** Do you mean soil organic matter \(SOM\) content?
**Response to AEC16:** Revised L318

**AEC17:** isotopic composition of soil water.

165 **Response to AEC17:** Revised. L322

**AEC18:** soil depth
**Response to AEC18:** Revised. L322

170 **AEC19:** to what?
**Response to AEC19:** Calibration was not sensitive to the initial conditions. The authors have clarified this in revision. L322-323

**AEC20:** Please describe explicitly which variables were used to force the model.

175 **Response to AEC20:** The forcing data were all explicitly listed in Table 2. The authors have added a cross-reference to the table in the revision to improve clarity.

**AEC21:** But if you use all available data for model parameterization, the model performance must necessarily good. The better way would be to split the dataset in a training/parameterization
180 and a validation part.
**Response to AEC21:** This is not necessarily true, different calibration data sets can "pull" the model outputs in different ways, and multi-criteria calibration is often used as an exacting test of model skill. Consequently, simple splitting of calibration and validation periods used in traditional rainfall-runoff models is not always appropriate. Moreover, since a model calibration
185 best represents the calibration period that was used, calibrating to only the primary growing period when latent heat and water usage is high could limit the usefulness of the information provided during a validation period where conditions (not remotely close) to calibration are experienced. While the information content gained from extrapolating to unobserved conditions is useful for forecasting how models predict future conditions, it was not the goal of this
190 manuscript. Furthermore, a model used for predictive a tool would typically already use at minimum a full year (seasonal cycle of radiation) prior to validation. As we do not have this length of data (given the resource-intensive nature of the in situ monitoring), this data split becomes impractical. The authors have added a further justification of the calibration and soft validation approach used in this study. Section 3.4.2.

195

**AEC22:** soil layer
**Response to AEC22:** Revised. L359

**AEC23:** Into which depth slices did you sub-discretize the first layer?
200 **Response to AEC23:** The subdiscretization of the soil layers was already described in Section 3.2.2. The authors have added this to the results section to improve clarity L362

**AEC24:** moisture values
**Response to AEC24:** Revised L364

205

**AEC25:** Do you mean soil water depletion \(= drought\) or soil water isotopic depletion?
**Response to AEC25:** The authors have clarified that this referred to soil water isotopic depletion. L368

**AEC26:** Sloppy expression. Should read like "...with the soil water isotopic values at greater depth \(100 cm\)...."
**Response to AEC26:** Revised. L371

**AEC27:** the measured values
**Response to AEC27:** Revised. L371

**AEC28:** strong? pronounced?
**Response to AEC28:** Revised L373

**AEC29:** Was
**Response to AEC29:** Revised. L381

**AEC30:** noticeably or clearly
**Response to AEC30:** Revised. L374

**AEC31:** But isn't that a sign of insufficient process representation in the model? In other words, wouldn't a better model performance at smaller time steps indicate a better representation of the processes?
**Response to AEC31:** This is a very common issue in spatio-temporal scaling, whereby changes in scale help to reveal model limitations and future considerations for improving modelling. The authors have already included a discussion of model cell storage contributions. We presented multiple temporal scales to help reveal how well the approach could estimate this variability and have discussed reasons why the model was unable to reproduce the large variability observed in the xylem. As indicated in the section, the variability in xylem water isotopes was much larger than the variability of measured soil water isotopes which is why even instantaneous mixing of isotopes was unable to reproduce the variability observed in the xylem isotopes.

**AEC32:** those? To which noun does "the" refer to?
**Response to AEC32:** Revised to "those" L383

**AEC33:** water.
**Response to AEC33**: Revised L386

**AEC34:** Sentence not understandable.
**Response to AEC34:** The sentence has been revised. L507

**AEC35:** Please specify the expected ranges. They can vary substantially between studies.
**Response to AEC35**: The authors have added the expected ranges to the section (L529). Our ranges are presented in the result section.

**AEC36:** See previous comment. Please specify the Tr/ET fraction simulated with your model compared to other studies.
**Response to AEC36**: As with **Response to AEC35** the authors have added the expected Tr/ET ratio to the discussion. L529

**AEC37:** But where did you get the required soil moisture profiles for the adjacent model cells from? Or was the assigned capability of the willows of using water from adjacent cells simply a way to "close a gap" in your model results?

**Response to AEC37**: The data from the grassland monitoring site was used as representative of
260  the surrounding area. The authors have revised the methods section to improve the description of the water sourcing developed for this study. Section 3.2.1. This isn't "closing a gap" it's using the model-data fusion as a learning framework to identify where the model needs improvement. This development is an advance that has wider implications for future model application (e.g. at the boundary between forest/non-forest areas where rooting may extend beyond the edge of a
265  canopy).

**AEC38:** Noticeable
**Response to AEC38:** Revised. L546

270  **AEC39** You mean from groundwater at all, right?
**Response to AEC39:** Revised. L546

**AEC40:** Or simply due to the differences in plant-functional type \(tree vs. grass\)?
**Response to AEC40:** The authors have revised the statement to include this suggestion as a
275  possible mechanism for the rooting distributions. L551-552

**AEC41:** Awkward sentence, not understandable.
**Response to AEC41:** The sentence has been revised to improve clarity. L560-563

280  **AEC42:** Again, please specify how you dicretized the soil layer, i.e. in which sublayers.
**Response to AEC42**: As with **Response to AEC23**, further clarification of the discretization of soil layer 1 has been added to the results section, though this was already presented in the methods section.

285  **AEC43:** Please elaborate on how this horizontal and vertical heterogeneity might have impacted your model performance, and more importantly, the representativeness of your study site.
**Response to AEC43:** The authors have added further discussion of the potential impacts that further measurements may have on the model performance. L574-580

290  **AEC44:** You mean the variability of the waters' isotopic composition, right? Could also be, e.g., the water potential. Please add.
**Response to AEC44:** Revised. L583

**AEC45:** isotopic composition of xylem water was
295  **Response to AEC45:** Revised. L593

**AEC46:** How can the \(measured? modeled?\) seasonal magnitudes of xylem water isotope dynamics be predominantly due to differences in simulated vs. measured soil water isotopic composition? I don't understand the reasoning here.
300  **Response to AEC46:** The simulated seasonal dynamics of xylem water isotopes had periodic differences from the measured seasonal dynamics of xylem water isotopes. These periodic

differences were noticeable when comparing xylem water isotope estimation using simulated or measured soil isotopes. Seasonal dynamics improved when measured soil isotopes were used because the measured source water is more accurate for mixing than "estimated" source water. The authors have clarified the statement and have provided a moving average of measured xylem water isotopes to Fig. 6 to better show these dynamics. L596-600

**AEC47:** Could this mean that modeling becomes dispensable?
**Response to AEC47:** No, this statement indicates that for our study the isotope data collected were useful datasets for calibration and assessment based on the integration of energy and water balance and biomass production. Modelling approaches yield large amounts of data and insights including time series of variables (e.g. water ages) that can not be directly measured.

**AEC48:** But this means that essential processes are not \(sufficiently\) covered by the model.
**Response to AEC48:** We would argue that virtually all models have limitations and identifying them is a pre-requisite to model improvement. The remainder of the discussion section already describes potential reasons for why the model was unable to fully capture this variability. The authors discuss measurement uncertainty (e.g. condensation, diffusion, and mixing), processes (e.g. fractionation), as well as model structure limitations. As per Reviewer 1 suggestions, the authors have added to the model structure section to include suggestions for further developments of the model.

**AEC49:** Tubing
**Response to AEC49:** Revised. L613

**AEC50:** from either of
**Response to AEC50**:Revised. L646

**AEC51:** Could you give an estimate of the maximum volume of cell water storage of the trees vs. available soil moisture?
**Response to AEC51:** The authors have included an estimate of cell water storage using the approximation provided by Čermák et al., 2007. L666

**AEC52:**high temporal
**Response to AEC52:** Revised. L677

**AEC53:** Either "throughout one summer" or "throughout the growing season".
**Response to AEC53:** Revised. L678

**AEC54:** Please specify how well the model captured the different variables.
**Response to AEC54:** Revised. L678-680

**AEC55:** If the measured data are indispensable for the process-based modelling, could they make the process-based modeling dispensable by using simple mixing models and Bayesian root uptake modeling?
**Response to AEC55:** The measured data are indispensable for the calibration to ensure that a suite of coupled physical and biological processes are adequately represented. As the goal of this

study was not only to estimate the xylem water mixing but to additionally characterise processes (e.g. biomass productivity, the evolution of water age distributions etc.) that cannot be estimated via simple mixing models or Bayesian uptake modelling. Consequently, we would argue that process-based modelling is not dispensable.

**AEC56:** Awkward sentence. Please reword.
**Response to AEC56:** The sentence has been revised. L685

**General Comments: Reviewer 1**

The rationale for the distance-based mixing development is to mimic the capacity of the root system to tap water pools at various depths and that may be laterally distant, in a spatially- and time-explicit manner. The authors take good care in considering the time domain, and describe in Sect. 3.3.2 how the non-zero length of the root system translates into root-scale transit times distributions. In the spatial domain, it seems that the modelling approach links xylem water to same-pixel (6x6 m²) soil water, both in terms of root uptake and signature (isotopic content or ages). My understanding is that transpiration in ECH2O-iso uses same-pixel water content, and the distance-based mixing application makes no clear mention of which simulation pixel is considered. Section 3.2.1 mentions that the proportion of "potential root-uptake from outside model cells containing vegetation", I find it confusing that no explicit mention of how this is actually taken account is further made, and Fig. 6 suggests that same-pixel signatures (soil and xylem) are compared. However, it is clearly stated in the Discussion that "small-scale [vertical] variations, as well as the large spatial differences from the soils below the willows and below the grass, and between different soil layers (Fig 3) reveal the significant heterogeneity of the site despite relatively immature soils and the local spatial scales" (L505-507) and then, crucially, that "around half of the [water] uptake (by root length and water availability) estimated to occur outside of the willow [pixels]"(L518-520). It is then likely that a significant part of the isotopic signal found in the xylem of Willow 2 originates from water pools in neighboring, dynamically-distinct vegetation patches, in particular the grass patch. It makes it difficult to then assess the added value of this "distance-based mixing" model, which seems to essentially add a lag-based component to water mixing in along the root-stem continuum, while fine-scale spatial patterns may play a crucial role.

This inference is only based on the main text though, as the source code for root-stem mixing does not seem to be part of the main EcH2O-iso repository referenced in this manuscript (if that is correct, it would seem appropriate that the authors publish the full source code used in this study). As such, this approach ressembles a conceptualzation adopted in an earlier study published by some of the authors, cited in this manuscript, where a tree storage component was shown to improve modelled xylem isotopic signature at a coarser spatial scale where lateral contributions may cancel out (Knighton et al., 2020).

Given the above, I encourage the authors to clarify throughout the manuscript what water pools (in particular, "laterally" speaking) are considered when quantifying root water uptake and associated isotopic signatures and transit times. If these are indeed limited to the local (same-pixel) scale, then the scope of this paper becomes more limited, and I suggest to discuss much more thoroughly the limitations of this study, beyond merely stating "the potential influence of spatio-temporal variability of source waters on xylem isotopes" (L.520-521), including a potential rejection of the adopted root-xylem model conceptualization.

Non-exclusively, a stronger case for the development of the distance-based mixing approach could be made using a case where the contribution of soil pools within root radial extent are considered in calculating xylem water ages and isotopic signatures (e.g. extrapolating from grass-patches values, since Landgraf et al. (2021) suggest that Willow 2 is surrounded by Willow1 and grass patches otherwise?). Ideally the water fluxes should also be factored in when

calculating transpiration ;if it requires a heavier development of the ECH2O-iso code, the associated limitation should again be thoroughly discussed, as a bare minimum.

The general concern described above also arose because it does not seem that the "distance-based" model significantly outperforms the default "instant mixing" approach (Figure 6 and Table 4), contrary to what is stated in the Discussion (L515-516). In evaluating the two mixing approaches, the authors took a very welcome step in comparing, in both approaches, the cases where "transit time and xylem isotopes were calibrated 1) using modelled soil isotopic compositions and sap flow, and 2) using measured soil isotopes and sap flow" as "The use of measured soil isotopes and sap flow tests the maximum potential for how each model performs and is not limited to the performance of EcH2O-iso for sap flow or soil isotope" (L273-276). In the end, I can only agree with the authors that "seasonal magnitudes of xylem isotope dynamics were predominantly due to differences in simulated v. measured soil isotopes in the shallow soils [rather than differences between mixing approaches]" (L527-528), and it also seems that AIC and KGE values, in the case of using measured soil istopes and sap flow, are rather close between "instantaneously mixing" and "distance-mixing" cases, with even KGE values slightly higher in the former case (Table 4). On a side note, it seems somewhat surprising that these higher KGE values translate into slightly worse AIC values given that the "distance-mixing" requires 4 additional parameters as compared to the "instantaneously mixing".

**Response to General Comments: Reviewer 1**

The authors thank the reviewer for their detailed feedback on the manuscript, which has aided in clarifying the methods presented in this manuscript as well as the interpretation and discussion of the results. During the revision, the authors have clarified that the model was developed in this study to allow for water usage outside of the current model cell using the new lateral root distribution component. This includes the description that water fluxes are considered from off-cell water storage and have provided the assumptions of water storage for off-cell contributions that are outside the model domain. The authors have further clarified that the xylem isotopic water and water ages are a weighted-average of within and off-cell root water contributions.

We have also clarified the significance of the distance-based model over the instant mixing model through further discussion of the efficiency criteria, the inclusion of a moving average of the measured xylem water isotopes to show the seasonality better, and a revised presentation of the results to aid in their interpretation. As the reviewer has mentioned, the KGE values are occasionally higher for the instant mixing approach compared to the distance-based mixing. The authors have improved the results section to clarify that the higher KGE is an artefact of the variability which "overshadows" the higher correlation coefficient and better bias presented by the distance-based approach. In this case the KGE "over-awarded" the instant mixing for variability that was not correct. These differences can be observed more clearly with the comparison to the moving average. Further, while the AIC and KGE appear "rather close", the difference in AIC directly presents a significance regardless of amount. To further show this, the authors have also included the standard deviation of the AIC and KGE which reveal that even if the absolute value of the AIC is not considered significant, the AIC of distance-based mixing is always significantly lower than the instant mixing.

Lastly, the authors have revised the manuscript according to the specific comments provided by the reviewer as detailed below.

**Specific Comments: Reviewer 1**

**R1C1: L31:** The 80-90% T/ET estimate by Jasechko et al. (2013) is often thought to be overestimated ; maybe the "updated" estimate Schlesinger & Jasechko (2014) would be more appropriate for citation.
**Response to R1C1:** The authors have revised the reference to Schlesinger & Jasechko (2014). L33

**R1C2: L36:** Please considering citing the original, peer-reviewed publication by Zink et al. (2017)
**Response to R1C2:** The authors have revised the reference to Zink et al. (2017). L36

**R1C3: L36-37:** I am not sure what is meant by "beyond vegetation uptake during the growing season", please rephrase.
**Response to R1C3:** The authors have revised this statement. L36-37

**R1C4: L43:** Rather than "small or larger scales", please consider providing indicative scale (e.g. plot to stand).
**Response to R1C4:** The authors have revised "small or large scales" to "plot or stand**.**" L43

**R1C5: L65:** Appropriate citations of ecohydrological modelling advances may also include Maneta et al. (2013) and Fatichi et al. (2012).
**Response to R1C5:** The authors have added these references. L64-65

**R1C6: L81-82:** The stated achievements are rather general; additionnally it would preferable to have this section turned this into research questions and/or testable hypotheses (it is not clear to me what these are), to further detail the general goal described L79-80. In this process, rather than "exploring" achievements/question #2 should better state the adopted stategy regarding root-mixing development and its evaluation/rejection (see General Comments)
**Response to R1C6:** The authors have revised the objectives to reduce the generality and improve the clarity of the overall objectives and research questions.

**R1C7: Fig. 1:** In connexion with the General Comments regarding the rooting system, it would be welcome to have a visual description of the land patches neighbouring the study plots (e.g. in Fig 1b or c, as in Fig. 1c in Landgraf et al., 2021), since the main text (L90-91) only describes what is at least 20m away from the plots.
**Response to R1C7:** The authors have revised Fig 1 to show the surrounding landuse patches in subplot c.

**R1C8: L117:** Did the author mean "Köppen Index Cfb"?
**Response to R1C8:** The authors have revised the climate index classification name to Köppen Index. L121

**R1C9: L147-155:** I could not find a description of how in-situ LAI measurements are carried out, although such data is presented in Fig. 5, could the authors clarify?

**Response to R1C9:** The authors have added the description of the LAI measurements in addition to the MODIS datasets to the materials and methods section. L142-144 and L329-333

**R1C10: L177-183:** It seems from the text that the version of code used in this study uses the SPAC module developed by Simeone et al. (2019), if so the authors should acknowledge and cite this work

**Response to R1C10:** The SPAC module was incorporated to test the vegetation and hydrological conditions, but was not found to show significant water limitations. As this was not a primary finding and the SPAC module was not used during the final calibration, this has been moved to the supplementary material to avoid confusion. The authors have included the citation (Simeone et al. (2019)).

**R1C11: L206-207:** Is it a full mixing in the whole soil domain? Or some compartments are differentiated?

**Response to R1C11:** Full mixing is conducted within each soil layer (10, 30, and 60cm depths) and within the canopy and surface stored water (when applicable). The authors have revised the statement to better clarify the isotope and water mixing in the soil. L216-219

**R1C12: L217:** The 100 "best" simulations have not been defined yet, please refer to Sect. 3.4.2

**Response to R1C12:** To improve clarify the authors have removed references to "best simulations" and have revised the statement to indicate "parameters sets were re-run". An additional reference has been added to indicate that this method have previously been used. L229

**R1C13: L240:** I do not understand the synchrony between the proposed descriptino of rooting length and SPAC, as the latter module is mostly focused on tree mortality (roots included).

**Response to R1C13:** This statement was intended to describe the connection between the rooting distribution (vertical only) already present within EcH2O and the proposed lateral root distribution. The authors have removed the reference to the SPAC module to improve clarity.

**R1C14: Eq. (1):** I am not sure how this equation was derived from Sperry et al. (2016). I am guessing it combines the cumulative root proportion provided in Eq. (6) in the above reference, the use of center-of-biomass depth, and layer depths in EcH2O-iso, but the intermediate steps to Eq. (1) escape me. In addition, I am confused so as if the beta factor here is the same beta found in Sperry et al. (2016) and its relation to the exponential factor kroot, also because the value of 0.995 is also found (for beta) in Sperry et al. (2016) Also, in calculating the vertical length, shouldn't one add the height-above-ground at which xylem measurement are made (here, 1 meter)?

**Response to R1C15:** Equation 1 presented in the manuscript is a modification of the source code provided by Sperry et al. (2016). The authors have added the reference to the source code to the manuscript and have clarified how the current formulation has been modified to help improve clarity. The $\beta$ factor has an equivalent value here as in Sperry et al., 2016; however, the $\beta$ factor used in this study is estimated using the $k_{root}$ parameter from EcH2O rather than an independently set parameter as in Sperry et al. (2016). The authors equated the translation of parameters (as indicated in the text) which was tested to ensure that equivalent values were

produced before Eq1 was implemented into EcH2O. The coefficient 0.995, indicating 99.5% root biomass has been described in the manuscript. L260

535

**R1C15: L246-253:** This approach differs from Sperry et al. (2016), where the volume of roots is calculated in the first layer, using radial length in the first layer, and then radial in others layers is estimated by assuming that each layer has the same volume of root. It is likely not the case here because layer depth is fixed but kroot seems to be calibrated and differs between simulations. So

540 I am guessing the authors used total root volume, implying that Eq. (2) uses total rooting depth (rather than $d_1$ as currently written) and then use Eq. (3) as a custom-made formula to reach the radial lengths in each layer?
**Response to R1C15**: The reviewer is correct that the volumes of roots in each layer are not equal. The authors have revised the statement to indicate that the equation was modified from

545 Sperry et al., 2016 by using the proportion of roots in each soil layer to adjust the rooting volume. As with Eq1, this modification was tested prior to implementation in EcH2O to ensure that if rooting proportions were equivalent in each soil layer the root volume in each layer was also equivalent. L268

550 **R1C16: L249:** According Sperry et al. (2016), D should be the maximum rooting depth, not the total soil depth.
**Response to R1C16:** Within EcH2O, the vegetation rooting depth is maximized at the maximum soil depth. There is no additional parameterisation to reduce maximum rooting depth. The authors have clarified this equivalence in the revision. L266

555

**R1C17: L252-263:** While the principles of root-length-based transit times is nicely described, it is quite furstrating not to see the calculated values for the rooting length (radial, vertical, total) in the results section or elsewhere in the manuscript. This could be a supplementary figure or table, at a minimum.

560 **Response to R1C17**: The authors have add the rooting lengths to the supplementary material (Table S5).

**R1C18: L264-265:** At first glance, this no-cavitation hypothesis seems inconsistent with the integration of the SPAC module, whose purpose is precisely to describe occurrence of cavitation

565 using plant hydraulics. Did the authors found evidence that no cavitation occurred during the simulated growing season?
**Response to R1C18**: Given the relatively dry soils below the willow trees, the authors recognised the potential for water stress characteristics which is why the SPAC module was incorporated during testing. Ultimately it was found that the willows were not under water stress

570 during the simulated growing season. As the approach utilizes sap velocity as an input time-series, some effects of cavitation would be experienced by the approach (e.g. decrease in flow rates); however, the approach does not account for other potential effects of cavitation along the flow path. The assumption was presented for the transparency of the approach. The authors have revised the statement to better clarify this assumption. L283-284

575

**R1C19: L288:** Do the authors mean that the bottom depth of each layer in the model is fixed to correspond to 10, 40, 100cm, with effective layer "thickness" of 10, 30, and 70 cm, respectively?

This information is provided in Table 3's caption, but it would be handy to have it earlier in the manuscript.

580 **Response to R1C19**: The individual measurement depths were already provided in the materials and method section (L140). As the depths of each soil layer are specific to the study site presented here, the introduction of layer depths prior to the model set-up section could create confusion on the model capabilities.

585 **R1C21: L293-295:** How is the grouping done for vegetation parameters? This is quite unclear, all the more that the type of information on calibrated parameters in Table S1 is not provided for vegetation parameters, could the authors provide a similar table? In addition, the SPAC module requires further parameterization that was carefully constrained in Simeone et al. (2019), but no mention is made on this topic, nor associated parameters, in the manuscript. Overall, it seriously 590 limits the reader's understanding of the modelling setup used in this study.

**Response to R1C21**: The authors have added the vegetation parameters to Table S1 in the supplementary material. Due to the availability of water from outside vegetation model cells, SPAC parameters were not sensitive for the final calibration and were not considered for analysis. As with **Response to R1C10,** the authors have moved the description of SPAC to the 595 supplementary material to improve clarity.

**R1C22: L305:** Have the authors looked at the additional information brought by lc-excess? This could further helps analyzing contribution from shallow/deep soil horizons, and further fractionation effects (or lack thereof) during root-stem transport.

600 **Response to R1C22**: The authors did simulate both $\delta^2 H$ and $\delta^{18}O$ and did not find large differences between the variables and measurements. The authors presented only $\delta^2 H$ as it was directly calibrated. The $\delta^{18}O$ did not produce notably different results (i.e. same dynamics relative to measurements). Furthermore, the sensitivity of 2H to fractionation is greater than 18O. Through the revision, the authors have clarified that isotopic soil datasets were used as an 605 additional dataset as part of a framework to evaluate water and energy fluxes simultaneously rather than a tracer-based experiment only. For further transparency of the results, the authors have included the soil $\delta^{18}O$ simulations in the supplementary material and have included goodness-of-fit of modelled lc-excess to measured (calculated) lc-excess.

Given the similarities in water contribution from shallow/deep soil horizons to Bayesian mixing 610 (Landgraf et al., 2021) the authors are confident in the model's capabilities to estimate the contribution with the current multicriteria calibration. It was not an objective of this manuscript to assess potential fractionation in the root-stem transport, and the transport was assumed to be non-fractionating. The authors have clarified this in the revision. Section 3.3.2 and Section 3.4.2

615 **R1C23: L307:** By "split calibration", do the authors mean using a calibration period and a validation period? Or a calibration period for one step, and another period for the other step? A combination of both? Please clarify.

**Response to R1C23:** The authors have removed this potentially confusing terminology from the manuscript to clarify the method and justification. Datasets were not temporally partitioned into 620 calibration and validation periods as the full length of the datasets were required to appropriately parameterize the water balance, energy, and vegetation (biomass) parameters. As the model was not used for predictive measurement, remaining datasets ($\delta^{18}O$ and soil temperature) were used in "soft" validation in addition to evaluation of the consistency of successful multi-criteria

calibration (e.g. root-uptake distributions). The authors have clarified this in the revision. L236-245

**R1C24: L309-315:** I am confused so as to how this step-wise calibration was performed. First, I am interpreting L309-310 as having a first step using isotopes, energy fluxes and water balance data as a constraint, and then a second step using biomass data ; or rather, 4 steps for each data group? Please be more explicit, and possibly add this information to Table 2 as well. Secondly, since each steps use 100,000 samples, I am guessing that step i+1 does not use a subsampling from calibration step i ; how were the calibration steps connected? Overall, this section needs a substantial rewriting to understand how calibration was actually performed ; under which hyptoheses regarding parameter space, total number of parameters, etc. Consider adding additional supplmentary tables with information on calibration ranges at each step, resampling procedures, etc.

**Response to R1C24**: The authors have revised Section 3.4.2 to clarify the stepwise calibration. The stepwise calibration was conducted in two steps: first with the isotopes, energy, and water balances, and second with the biomass data. Because the LAI time-series was used in the first calibration step, the influence of biomass simulations on the first step calibration was negligible. To calibrate only for biomass, resampling of first step calibration parameter sets was conducted. The "best 100" simulations show both independent soil and vegetation parameters. L240-245

**R1C25: L325-328:** How was the sub-discretization done? Also, why not trying to change the thickness of the first layer so that the measurement depths fall withinthe model layers, not at interfaces between model layers (e.g. layer 1 could be 20cm-deep)? Adding the same red line to L1 mositure under grassland could be informative in checking for percolation ; from these figures it seems that infiltration-percolation under the grass patch is underestimated.

**Response to R1C25**: The subdiscretization was conducted using the soil water redistribution routine within EcH2O at 1cm soil depth increments, as described on L197-199 (initial submission). The authors have reiterated this description in the results section during the revision (L362). Changing the thickness of the soil layers to have the midpoint as the measurement location would result in unintended effects on the water and energy balance. Increased depth to soil layers dampens both the soil moisture and soil isotope responsiveness due to larger volumes for water movement and mixing. While the added depth in layer 1 may result in increased soil evaporation and some increase in fractionation potential of soil isotopes (particularly at site A), the added volume for mixing would reduce the total fractionation within the whole soil layer. As described in the manuscript, the discretization was not utilized in calibration, thereby the "damped" soil moisture response with a deeper soil layer would greatly impact calibration (L197-199 in initial submission). The authors did additionally sub-discretize the grass site, but due to wetter conditions and more frequent percolation (as shown by the more dynamic moisture in layer 2) the discretized moisture at 10cm was not notably different to the average of the soil layer. Infiltration/percolation appears likely to appear low due to the underestimation of the soil moisture in layer at the grass site. It should be noted that the parameterisation of soil layers is uniform, thereby multiple layers must be estimated with the same parameter set.

**R1C26: L330:** Another obvious isotopic feature is the much higher ~week-scale variability in 10cm isotopic at site A (Fig 3a) as compared to site B (Fig 3b) . This is reasonably differentiated in the simulations cells although 1. simulations at site A are too dampened and 2. there an

670      unrealistic depletion in October at site B. While the former is briefly mentioned in Sect 4.3, I
suggest to add these descriptions here and discuss them further on in the Discussion.
**Response to R1C26**: As described in **Response to R1C25**, part of the reason for more damped
simulated isotopic compositions at Site A is in the mixing within the soil depth. A secondary
part, particularly in day-to-day variability is soil-vapour interactions. Modelled isotopic
675      variability is dependent on infiltration and soil evaporation only. At Site A, the subdaily
variability is quite large as shown with the revised Figure 3. Thereby, variability in soil isotopes,
particularly depletion when no infiltration occurs, cannot be estimated by the model. The model
estimates the averaged isotopic conditions. Increases in the modelled isotopic variability are
likely only feasible through model structure development (outside of the scope of this study) or
680      through decreasing the soil layer depth. However, decreasing soil depth would decrease soil
evaporation. While there is an "unrealistic" depletion of soil isotopes in October at Site B, this
depletion occurs notably at site A due to depleted precipitation isotopes. The difference between
sites is likely due to differences in mixing. The authors have added additional descriptions of
these characteristics to the discussion. Section 5.1

685

**R1C27: Figure 3:** Are isotopic datasets daily-averaged in this figure? If so, it should be stated
somewhere in the main text.
**Response to R1C27**: The isotope datasets were daily averaged for visual purposes. The authors
have revised the figure and figure caption.

690

**R1C28: L344:** The model description states that there are two thermal layers in EcH2O-iso
(without providing the depth of each), can the authors briefly describe how they extrapolated the
modelled soil temperatures at three depths?
**Response to R1C28:** The soil temperatures at different depths were estimated using the same
695      linear damping formulation used to estimate soil temperature at the bottom of the thermal layers
(Maneta et al., 2013). Different depths were estimated by using soil depth of different layers in
the formulation following the estimation of surface temperature and thereby are not accounted in
the energy balance. The authors have added the description of the soil temperature estimation for
each soil layer to Section 3.2.1.

700

**R1C29: L345:** Although the scales in Fig. 4 (Site B, latent & sensible heat fluxes) are quite
squeezed (please consider expanding them), it is apparent that latent heat is overestimated
thoughout the growing season.
**Response to R1C29:** The authors have modified Figure 4 to better show the scales of the energy
705      balance components, and present the uncertainty of both simulations and measurements. The
revised figure better shows that there is not a systematic over-estimation of the latent heat
through the growing season at the grass site.

**R1C30: Figure 4:** How was modelled grass transpiration converted into sapflow? It would be
710      informative to see the transpiration rate (mm/d) in the second row, perhaps using a secondary y-
axis on the right?
**Response to R1C30:** The authors have revised the y axis label to indicate that this is the
volumetric water utilized by the grass. To keep time-series data more visible for interpretation,
the authors have included the transpiration rates in the supplementary material and have referred
715      to the relevant figure in the Figure 4 caption.

**R1C31: L358-364:** Could the authors precise which MODIS LAI product was used? These products usually have a much larger spatial resolution (500m-1km) then the modelled domain of this study. Can the authors develop on their methodology and assumptions made to distinguish willow and grass patches?

**Response to R1C31**: The MODIS dataset used was MOD15A2, the authors have added the description of the MODIS products to section 3.4.2. As the reviewer mentions the large (500m) resolution does not provide a clear division of vegetation types. The area surrounding the large (500m) area surrounding the site encompasses a greater majority of leafy tree vegetation, resulting in LAI from MODIS representing LAI of willows more than the grass. As downscaling this information is complex, the authors scaled the LAI dynamics for the grass as done in Smith et al., (2021). The authors have added this description to Section 3.4.2.

**R1C32: L370:** A reference to Table 3 would be useful.

**Response to R1C32**: The authors have added a reference to Table 3.

**R1C33: Table 3:** This table shows a lot of information. It might be much more reader-friendly if transformed to a multi-panel plot, either using bar or points with errorbar, e.g. keeping the row and column organization with facets and a color code for time periods. In addition, the third grouped-row (RU-L*) might be more intuitive if instead of layer number, depth ranges were used (e.g. RU[0-10cm]).

**Response to R1C33**: The authors considered translating Table 3 to a Figure; however, we feel that the loss of detailed information on water balance and biomass allocation provided by the Table would be detrimental to the interpretation and transparency of the results. Therefore the authors have retained the Table in the revision. As suggested by the reviewer, the authors have changed RU-L* to RU[depth].

**R1C34: Figure 6:** My understanding is that soil isotopes are measured in-situ at three depths, as reported in Fig. 3; why then are there not 3 solid lines in the diurnal plots, instead of 1 (panels a) and c)) or none (panels b) and d)), and why is the solid (measurement) line flat, as if there no high-frequency information? Additionnally, given the high-frequency dynamics, readability would be improved by making this figure wider, e.g. having Willow 1 and Willow 2 panels on top of each other.

**Response to R1C34**:The authors thank the reviewer for identifying this error. The soil legend has been corrected to indicate that measured soil isotopes are dashed lines and solid lines are simulated soil. The model does not estimate significant sub-daily variability in soil isotopes. All three soil depths are present on the plot; however, the simulations are all non-diurnal (overlapping). The authors have revised Figure 6 to widen the plots and additionally have shown the moving average of the measured xylem to aid in the interpretation of the variability.

**R1C35: L412:** A reference to Fig 3a (in addition to Fig 6a & b) would be helpful.

**Response to R1C35**: The authors have added a reference to Figure 3. L452

**R1C36: Table 4:** I am assuming the values between brackets give KGE variability among best runs? If so, why isn't the same number given for AIC? Consider using a plot rather than a table (although less critical than for Table 3).

**Response to R1C36**: The values in brackets for KGE (as indicated in the caption) are for sub-daily variability. To help clarify the authors have revised the caption. The authors have additionally added the standard deviation of AIC and KGE of the simulations. Evaluation of only sub-daily variability for AIC was too short for significant testing which is why they are not shown. A plot of this data would be more difficult to depict without many subpanels because the scale of AIC changes for different time-steps.

**R1C37: L440-449:** In my view this labelling by "contributing month of the year to current store/flux" rather provides a very nice perspective, equally important and intuitive as the "time elapsed since arrival" reported above ; it directly replies to the question "what precipitation period is most important for plant water use?" ; I would suggest to move key Fig. S3 to the main text.
**Response to R1C37**: The authors have revised Figure 7 to show the monthly proportion of root uptake as suggested and have moved the fractional proportions to the supplementary material.

**R1C38: Fig. 7:** "Time in xylem" (panel g) is somewhat misleading, as the transit time considered integrate transit along root and xylem? Besides, my impression was that transit length (and thus time) in the xylem was neglected when computing v(i) in Eq. (1) (see related comments above)?
**Response to R1C38**: The authors have revised the y-label to "transport time in vegetation". Eq. (1) provides the vertical distance of biomass and Eq. (3) provides the radial distance. As indicated in Fig. 2, this translates to the distribution of transit times as a function of root length. Longer rooting lengths produce a longer transit time.

**R1C39: L450:** "an incrase of zero days" seems somewhat odd, maybe rephrase: "Since intantenous(ly?) mixing equates xylem water age to that of where water is taken up (reaches 1m instantly), transit times along the root-xylem system are only shown…".
**Response to R1C39**: The authors have revised the statement. L507

**R1C40: L479-496:** The underestimation in modelled willow transpiration (or rather, th sapflow, see a comment above) at the end of the growing season is quite interesting, as perhaps not as "minor" as stated here ; the model-data discrepancy exceeds the dispersion among best runs. That would deserve further discussion, as the current ones somewhat circumvent the issue with more general considerations. Besides, the concommitent overestimation of modelled L1 moisture (and possibly L2's, and thus percolation, Fig. 3a) suggests that it's not necessarily due to missed contributions from adjacent cells or a short-term reliance on deeper stores (which would have been interesting as a drought-protection process!), but merely that there is something wrong with evaporative demand when the energy balance is computed ; is it something due modelled energy fluxes and/or to forcings? In other words, is a process being missed?
**Response to R1C40**: The authors have added further context to the discussion regarding the under/overestimation of fluxes. The authors have highlighted two reasons for the discrepancies, firstly the non-temporal variability of lateral root-uptake (i.e. higher near stem uptake with water available), and secondly parameterisation. While adjustments in parameterisation could aid in reducing the discrepancies it comes at the cost of parameter feasibility. Section 5.1

**R1C41: L480-481:** Is this sentence suggesting that EcH2O-iso account for off-cell contribution to calculate root water uptake? And associated transit times?

**Response to R1C41**: The modifications made to EcH2O-iso calculate the proportion of vegetation water utilized within and outside of the cell of the vegetation. Thereby where water was used from outside the cell, water age and isotopes were additionally considered. Where model domain was exceeded, moisture and isotopes were assumed to be equivalent to the grass as it surrounds the model domain. Mixing in the transit times consider water source (within v. outside cell). The authors have revised the methods section to clarify the modifications to EcH2O to allow for off-cell contributions. L187-191.

**R1C42: L492-494:** From the 'slight descrease' I am wondering if the authors meant "was under stress"? Besides, it could be informative to further have the absolute biomass in each compartment (in addition to biomass allocation) reported somewhere, perhaps as time series over the growing season, to check if the potential decay rates exceed (or not) allocation, and where.

**Response to R1C42**: The authors have clarified in the revision that the decrease in root biomass production (relative to leaf and stem) during the growing season is consistent with a willow not under water stress L555-557. The vegetation allocates more biomass production to leaves and stems. If the vegetation were under water stress, root biomass production would increase as the vegetation "searches" for water. In light of the length of the manuscript and supplementary material and as the growth and decay of biomass components is already inherently shown with LAI and stem growth, and with GPP directly related to transpiration, the authors have not included the biomass time-series.

**R1C43: L514-521:** I assume this part of the discussion will be substantially revised (see General Comments)

**Response to R1C43**: As the authors used the AIC to directly inform on the significance of adding parameterisation to better understand the xylem isotope dynamics, it is unclear why a potential rejection of the modelling approach should be considered. To aid in the clarification of the significance of the approach, the authors have revised the results and discussion section to provide further insights on the "discrepancy" provided by the KGE. The KGE of the instantaneous mixing KGE was "artificially" high due to the much greater variability where instantaneous "peaks" did not correspond to measured "peaks". Consideration of correlation coefficient and ratio of means were both greater for the distance-based approach than the instantaneous mixing. Furthermore, the authors have clarified that the instant-mixing assumption failed to capture the dynamics of diurnal variability which suggests missing processes from the model. (L. 450-451 and 586-588)

**R1C44: L536-539:** If the measurement uncertainty is known, it would be highly informative to add it as error bars on any related plots presented in this manuscript. Actually, it should be common practice, helping to tamper interpretations where inferred dynamics are commensurate with uncertainties.

**Response to R1C45**: Error bars related to vegetation isotopes were already presented in Fig. 6, which highlights the wide ranges of xylem isotopes. The authors have added error bars to the sapflow, and soil moisture, temperature and isotopes to better present the datasets.

**R1C45: L546-553:** This issue could be explored with the different tree storage mixing types presented in Knighton et al. (2020), it could help the current discussion and open avenues for further development?

855 **Response to R1C45**: The authors agree that a combination of tree storage mixing and root-stem transit mixing presents open avenues for further development. The authors have added brief discussion on potential model developments. L618-628

**R1C46: L552:** Are the authors referring to measured basal diameter?

860 **Response to R1C46**: This has been revised. L627

**R1C47: L561-563:** Maybe further precise "across Switzerland" after "Allen et al. (2019)" and "in the study region" before "(Miguez-Macho and Fan, 2021)"?
**Response to R1C47**: The authors have revised the statement according to the reviewer's

865 suggestions. L635-639

**R1C48: L575-578:** This is also potentially due to the fact that other studies considered root-to-shoot transit times (Meinzer et al., 2006) while this study "stops" at 1m height.
**Response to R1C48**: While this will have some influence on the transit time, the velocity in the

870 xylem is relatively fast, which even during the late growing season would only add ~3 days on average to the transit time to the leaf. Vegetation species properties contribute to greater changes to the transit time than measurement height in this study. The authors have revised the statement to indicate the potential differences due to measurement location. L650-652

875 **R1C49: L579:** Essential or indispensable?
**Response to R1C49**: The authors have revised the statement. L655

**R1C50: L580-591:** Again, it is quite surprising not to see any references to Knighton et al. (2020), a study the authors contributed to, and which precisely studies this issue of tree water

880 storage and mixing.
**Response to R1C50**: This section was intended to directly discuss the impact of cell storage release as a contribution to root-stem transport mechanisms. It was not the intention to downplay the significance of the results of the work conducted by Knighton et al. (2020) which is acknowledged elsewhere in the manuscript. As with **Response to R1C45**, the authors have

885 added further discussion of tree water mixing implications with reference to Knighton et al. (2020).

**R1C51: L589:** Mennekes et al. (2021) and Benettin et al. (2021) are recent studies on this topic, albeit in semi-controlled conditions.

890 **Response to R1C51**: The authors have added these references to the discussion. L668

**R1C52: Conclusion:** Having the Conclusion framed as Summary (L596-608) seems a bit redundant with the abstract and the main text. Rather, discussing high-level limitations, insights and potential avenues would more efficient.

895 **Response to R1C52**: The authors have revised the conclusions and reduced the "summary" of results to include more high-level interpretations.

**R1C53: Figure S1:** The channel is not represented in Fig 1b, and the similar color code for snowpack/channel may confuse the reader).

900     **Response to R1C54**: The authors have removed "channel" from the legend in Fig S1.

**R1C54: Code availability:** The statement is somewhat incomplete, as the post-processing model to compute root geometry and associated transit times does not seem to be on the referenced repository.

905     **Response to R1C54**: Upon publication, the authors will update the bitbucket code to better reflect the code version used in this manuscript.

**R1C55: Data availability:** Again, this statement is misleading, first because "open-access" is incompatible with password-protection. Secondly, not all the data used in this manuscript is

910     archived in the provided link ; only sapflow, stem variation and in-situ isotopes data are listed, while neither eddy-covariance energy fluxes, micrometeorological measurements, in situ LAI, and soil moisture can be found. I would strongly encourage to have all datasets published, or at a minimum have them listed along with their open-access metadata on FRED so that potential users can make informed queries to the curators.

915     **Response to R1C55**: Upon publication, the authors will update the data available on FRED to better reflect the data used within this manuscript.

**General Comments: Reviewer 2**

920 The study by Smith et al. (bg-2021-278) presents a novel combination of in situ temporally high-resolution measurements of micrometeorological variables, water fluxes, stores and stable isotopes in soil and xylem together with a process-based modelling approach, in order to identify the dynamics of water partitioning under 2 willow trees and a neighboring grass patch over a growing season. The increased perspective on soil-plant water dynamics brought by this

925 intensive monitoring, further presented in another manuscript (Landgraf et al., 2021) is used a for a multi-data calibration and evaluation of the ecohydrological outputs provided by the EcH2O-iso model. The authors use this baseline to then evaluate a new conceptualization of water uptake and transport along a vertically-and-laterally-distributed root profile, in order to understand the relation between soil and xylem water dynamics and signatures.

930 The topic of this study is highly relevant and timely. The ecohydrological community is 'on alert' at present, with novel opportunities arising from in situ, higher-frequency isotope measurements in soil and plants. At the same time many new discoveries related to methodological issues measuring water isotopes in these compartments arise steadily. Both aspects provide opportunities, but also a number of challenges related to modeling these datasets.

935 The presented study is a complex and well-conducted investigation on how to combine multi-facetted datasets into a joint modeling framework. This is certainly something I applaud the authors for. With multiple years conducting in situ isotope and ecohydrological measurements in several environments, it is simply great to see how such datasets can be put into one modeling framework. Having that said, I find it crucial to also implement measurement uncertainty into

940 modeling frameworks. All the recently discovered isotope effects certainly increased the measurement uncertainty, and this – in my opinion – also increases model uncertainties? Can we even make reliable quantitative statements considering both? I know this goes farther than this publication, but I think it is necessary to have this in mind. Hence, the way this modeling exercise was carried out is excellent and has great potential for using such models for other,

945 recently recorded, in situ datasets. However, the quantitative estimates of this study only incorporate modeling uncertainties. The lack of replication, uncertainty of soil and plant water isotope measurements, and spatial variability of ecohydrological measurements makes the quantitative value of the modeled data at least questionable. While it is probably impossible to address this in the presented study, this should certainly be on the future agenda. However, an

950 honest evaluation and interpretation of the modeled data in that regard would benefit the manuscript in my opinion. At the same time, the manuscript could be shortened by putting less emphasis on the quantitative results and more on the modeling framework, strengths and also weaknesses.

In summary, the study definitely deserves to be published in BG, but requires thorough revision.

955 **Response to General Comments: Reviewer 2**
The authors thank the reviewer for their thorough review of the manuscript, which has aided in improving the interpretation and significance of the methods and results presented in this manuscript. Through the revision, the authors have aimed to clarify the uncertainty and variability associated with the measurements and how the uncertainty relates to the model results

960 and model uncertainty. The general consensus of modelled and measured uncertainties in this

study help to better reveal the capabilites of the model approach. The authors have further aimed to clarify the measurement proceedures to improve the self-sufficient nature of the manuscript. Lastly, the authors have further revised the manuscript using the specific commens provided by the reviewer.

965 **Specific Comments: Reviewer 2**
**R2C1: l.75**: I would leave out importantly. It is important, but doesn't need this explicitly here
**Response to R2C1:** The authors have removed this statement.

**R2C2: Fig. 1:** Figure caption is incomplete, in particular d) what are the blue and red bars? What
970 is the grey box?
**Response to R2C2**: The authors have revised the figure caption in Fig 1d.

**R2C3: l.144**: Sensors were installed until 1 m soil depth. Is that the maximum rooting depth for both willow trees and grass? This is crucial for root water uptake depth determination
975 **Response to R2C3**: The authors have clarified in the revised manuscript that visual inspection of the rooting density during the installation of the soil moisture and soil vapour, a high proportion of the rooting density was observed in the shallow soils (L149). While the maximum rooting depth of each tree was not directly measured, further measurements of groundwater (~2.2m) were also taken, but it was determined that vegetation source water was not taken from
980 groundwater (Landgraf et al., 2021) and dominant root uptake depth from >50cm.

**R2C4: l. 145-160:** Even though I understand the method is described in Landgraf 2021, the information on how isotope standards were prepared and measured would be good here. Also, referencing the borehole method because of the short description herein should be considered.
985 **Response to R2C4:** For brevity of the manuscript., the authors have added a further description of the methods to the supplementary material. The authors have added the reference for the borehole methods to the methods section. L160

**R2C5: l. 178-180:** and chapt. 3.2.2: how were these parameters determined/calibrated?
990 **Response to R2C5**: The authors have added the parameter ranges of the vegetation parameters to the supplementary material. The parameters were determined through calibration, with ranges established utilizing previous studies in the region. The calibration procedure has been revised in the manuscript. Section 3.4.1

995 **R2C6: L.214:** the last part of the sentence is unclear, please rephrase and clarify
**Response to R2C6**: The authors have revised the statement. L227

**R2C7: L.216:** calibration? How was it calibrated?
**Response to R2C7**: The authors have revised the section to remove references to "calibration".
1000
**R2C8: L.216-223:** this approach is interesting, was this used somewhere before? (citation?). It appears like such an approach would completely neglect preferential flow, am I correct? If yes, this should be stated somewhere ('does not account for pref. flow')
**Response to R2C8**: The approach was used in Smith et al. (2020), the authors have added this
1005 reference to the section. This approach is dependent on the structure of the model applied (the

approach is not specific to EcH2O-iso). Since EcH2O-iso does not account for preferential flow, the results in this study will additionally not account for preferential flow. The authors have added a statement to the EcH2O water balance section to indicate that preferential flow is not considered. L229

1010

**R2C9: L.229:** assumed root distributions…this is a BIG assumption. How were they assumed?
**Response to R2C9**: As is common in ecohydrological models, the root distribution is estimated in EcH2O-iso using an exponential distribution for vertical and lateral rooting proportions. The vegetation water mixing routine used in this study does not explicitly estimate rooting

1015    distributions but uses distributions calibrated within the physically-based model. The authors have clarified this in revision. L253

**R2C10: 3.3.1.:** How were the root parameters determined/approximated?
**Response to R2C10**: As with **Response to R2C9** the root parameters are calibrated using
1020    transpiration (and sapflux) and isotopic measurements. Further descriptions of how rooting distributions were parameterized and utilized have been added to the manuscript. Section 3.4.1

**R2C11:** 3.3.2.: For someone who does not model every day, the explanation on root length determination should be clearer. Coming from the field side of things, I wonder 'how is
1025    maximum rooting depth implemented?'; which measured parameters does one actually need (precipitation and sap flow?). I also wonder, if the general root distribution in the model always has the same shape? This is a large simplification that is definitely not true for any given vegetation species. How does it look like if we have a deep-rooter, for instance?
How was the fact handled that there very likely were willow roots present underneath the grass,
1030    affecting soil water contents and hence, the modeling efforts?
**Response to R2C11**: The authors have expanded on the description of the parameterisation of the rooting distributions to Section 3.4.1. Evaluation of the final parameter values for root distributions, particularly lateral root extents is already in the Discussion (L589). All necessary data (forcing data) are presented in Table 2 for model set-up and running. Measuring additional
1035    variables (or parameters) is dependent on individual study sites, study objectives, and the sensitivity of the model to the output variable. It is outside of the scope of this manuscript to discuss all potential uses of measurement or model output variables. In revision, the authors have clarified that the maximum rooting depth is equivalent to the total soil depth (L266). For clarity, parameterisation of the rooting distribution permits rooting distributions to shift rooting
1040    distributions from shallow soil (Kroot > 5) to deep soil (Kroot < 1) with a smooth exponential shape. Of course, this is a simplification, but it is reasonable in the absence of further information to justify alternatives. The current model structure does not permit bimodal rooting distribution of deep and shallow soils (i.e. lowest proportion in layer 2). The model here was adjusted to allow for roots to occur from outside of each model cell. In this way, willow roots
1045    could access water below the grass. This has been clarified in the revision. L190

**R2C12: L.277:** this is an interesting point, but it should be noted that there is not only an error in simulating, but also measuring soil water isotopes. I am not saying that it should be, but is there a way to include this in such simulations?
1050    **Response to R2C12**: While outside of the scope for this manuscript, there are methods to account for measurement uncertainty of both forcing and calibration data within model results.

This is generally evaluated externally to the model (e.g. GLUE) and included within the uncertainty bounds. To improve the transparency of the model simulations with respect to isotopic variability, the authors have shown the daily isotopic variability in addition to the mean values shown in Figure 3.

**R2C13: L.288-291:** Maximum rooting depth is constrained to 100 cm. This needs to be proven/backed up. Stating another paper under review/discussion (here and in many other instances) is sort of cheating, to me. Root water uptake depths shift over a year and it cannot be assumed for the time of experiment (~3 months) that 100 cm max. rooting depth are a given. Please clarify this; I do believe the authors and a quick search tells me that willow trees are generally shallow rooted. However, another citation would help.
**Response to R2C13**: We are disappointed with the accusation of "cheating". Simply for issues of manuscript length in this modelling-focused paper, we referred to the openly available HESS-D paper by Landgraf et al. (2021) for measurement details. The authors have added further justification for the shallow rooting depth of the willows used within this study (e.g. they are ~12 years old and installation of soil monitoring equipment confirmed sparsity of roots at 1m). L310 and L149

**R2C14:L.305**: please explain thoroughly, why 18O was not used in calibration
**Response to R2C14**: Initial testing of model results did not reveal notably advantages to utilizing $\delta^{18}O$ within the multicriteria calibration with relative differences of simulated to measured $\delta^2H$ and simulated to measured $\delta^{18}O$ showing very similar responses. Using both isotopes in calibration could reduce the constraints imposed by other (non-isotopic) measurements, which would be inappropriate for a physically-based model. The authors retained $\delta^{18}O$ to help internally validate that the final calibration of $\delta^2H$ is not over-calibrated. The final $\delta^{18}O$ simulations are shown in the revised supplementary material.

**R2C15: L.306:** What is meant with 'the values for 18O were not greatly different from 2H'? First off, these values are usually very different. Second, the dual-isotope space provides an excellent way of validating the effect of kinetic fractionation. Third, I feel like a comparison of measured and modelled values in dual-isotope space would greatly benefit the trust in the model, apart from the statistical parameters.
**Response to R2C15:** The authors were referring to the trends of 18O and 2H showing very similar responses for model calibration rather than the absolute values (**Response to R2C14**). The dual-isotope space for comparison of measured to simulated isotopic data would potentially only reveal some under-enriched shallow soil water below the Willow (as already shown in Fig. 3) where soil evaporation was limited in the model by water availability. As shallow soil isotopes were only one component of the multicriteria calibration, further plotting of additional isotopic variables would not likely reveal more than information already presented within the manuscript. The authors have further added the time-series of $\delta^{18}O$ simulations to the supplementary material to aid in the interpretation of kinetic fractionation.

**R2C16: Table 2:** Calibration data: Why is only sap flow of 1 tree used? Likewise, Surface Temp and latent heat only from site B? This seems subjectively chosen and is not explained in the text.
**Response to R2C16:** The authors have revised the study site description to indicate where data were available and have updated the table caption to better clarify that the data presented in the

data are all available spatial locations. Surface temperature and latent heat were measured directed above the grass site (Site B) with the AWS (Fig. 1). Sapflow was an average of the sapflow of both willow sapflow (range of sapflow data will be presented in the results during revision) and as both willows experienced the same conditions calibrating both trees to the same sapflow was not deemed necessary.Section 3.4.2

**R2C17:** L.324: …starting from likely, it belongs to discussion
**Response to R2C17**: The authors have moved this description to the discussion.

**R2C18 Results**: the subjective phrase like 'adequate' or 'slightly different' should be backed up by some objective measures in the results section.
**Response to R2C18:** The authors have revised the results section to remove subjective phrasing and/or have added objective measures to the manuscript, with a full table of efficiency metrics included in the supplementary material. Section 4.

**R2C19 L.335-338**: Just to clarify: The heating cables were not put inside the soil profile, or were they? I am asking this because we did this mistake once in my group and it turned out the cables heated the surrounding soil, hence, producing a heating of the area around the soil gas probes and tdr probes. As a result, one would calibrate data on a totally non-representative dataset that is highly influenced by the heating cables and not representative for the stand.
**Response to R2C20**: The heated cables were not installed within the soil profile, but were installed from the installed membranes to the soil surface. In this way, the soil heat profile was not impacted by the heated cables.

**R2C20: Figure 3:** This looks nice indeed, in particular for Site B! However, I repeat my statement from before that the dual-isotope space allows for a more precise evaluation of model performance and further interpretations such as root water uptake depth or kinetic fractionation. Another thing: There is definitely an uncertainty in the in situ isotope measurements, which is almost never incorporated into modeling. However, modeling always incorporates uncertainty in calibration results. I find this odd and not necessarily correct.
**Response to R2C20:** The authors thank the reviewer for their positive feedback. The authors hold the opinion that with the number of data points presented and the large overlap, differences, particularly temporal, between the simulated and measured may not be as notable in dual-isotope space. The authors have added uncertainty and isotopic variability (sub-daily variability) to the figures to more explicitly reveal the model performance against measured datasets.

**R2C21:** The complete section 4.1 does not make use of any goodness-of-fit criteria and uses subjective and biased statements throughout. For instance, the calibrated sap flow data is judged as 'adequately captured by the model'. If I look at Fig.4 I (subjectively) see that the dynamics are OK (Site A) while the magnitudes are sometimes. For site B, there are no measured values for sap flow. This is not convincing to me. I strongly recommend adding goodness-of-fit criteria here.
**Response to R2C21**: As with the suggestion by the reviewer in **R2C18**, the authors have added goodness-of-fit criteria to the results section to better justify the fit of the model. Though to some extent, the choice of GOF criteria can be as subjective as qualitative terms that are used simply to make the text more readable.

1145     **R2C22 L.343**: 'quite' well…objective measure?
    **Response to R2C22**: The authors have added goodness-of-fit criteria to the results section and supplementary material.

    **R2C23 L.396**: simulated day-to-day variability could not reproduce the measured values
1150     **Response to R2C23**: The authors have revised this statement.

    **R2C24**: 4.3: I find this section well-written and less subjective/biased. The general dynamics are met, but it needs to be said that an offset of 10 in d2H is already a large deviation (in isotope space). Now is that because of a non-perfect model fit or, and I am sure that it also plays a role,
1155     uncertainty in the in situ measurements. I feel like including some statements/metrics in regard to the measurement part of the second paper submitted by the authors could benefit the interpretation here. I find the aspect of the time-steps quite interesting: Why temporal resolution do we actually need? In isotope-space, daily is already a great resolution.
    **Response to R2C24:** Well, no model fits are perfect. The authors have added a brief statement
1160     regarding the performance of the model and interpretation of the results. While there are deviations from the mean value (shown in Fig. 6 as 12-hour averages) the simulations generally fall within the minimum and maximum xylem isotopic range over the time-period. As samples were available every 2-hours (as per the methods section), simulations within this range suggest the dynamics were captured to a first approximation (as also shown with the efficiency criteria).
1165     L461-464

    **R2C25:** L.479/480: 'with only minor under-estimation of the transpiration in the willows toward the end of the growing season'…I do not agree that the deviation is minor (>50%) nor that the fit is great for the rest of the period. The dynamics fit, but the magnitudes often do not. And at site
1170     B, no comparison is provided.
    **Response to R2C25**: The authors have clarified potential reasons for the deviation of sapflow and soil moisture in the discussion section 5.1. These include model structure and parameterisation. The use of KGE rather than NSE emphasised dynamics (mean and variability) over the absolute value of individual events. Here, the absolute magnitudes were strongly
1175     dependent on the soil moisture conditions below the willows. No comments on the sapflow at Site B can be made because there were no measurements of sapflow in the grass.

---

## Referee Report (RR1)

The study by Smith et al. (bg-2021-278) presents a novel combination of in situ temporally high-resolution measurements of micrometeorological variables, water fluxes, stores and stable isotopes in soil and xylem together with a process-based modelling approach, in order to identify the dynamics of water partitioning under 2 willow trees and a neighboring grass patch over a growing season. The increased perspective on soil-plant water dynamics brought by this intensive monitoring, further presented in another manuscript (Landgraf et al., 2021) is used a for a multi-data calibration and evaluation of the ecohydrological outputs provided by the EcH2O-iso model. The authors use this baseline to then evaluate a new conceptualization of water uptake and transport along a vertically-and-laterally-distributed root profile, in order to understand the relation between soil and xylem water dynamics and signatures.

I thank the authors for providing a comprehensive and careful revised version of the manuscript that addresses many of the comments from version 1. I will first comment on the general points and then provide a few more comments that might help. Having that said, I recommend the manuscript for publication in HESS after addressing those very minor comments.

General points raised:

i.) I want to apologize for accusing the authors of 'cheating'. Probably, this came over too strong. I was simply wondering if it is completely justified to cite another paper that is under review. However, since HESSD is a citable source, this should not be a problem in this case.

ii.) Lack of replication: I thought about this again, and I think this should not be the concern for the modeling part of the study, i.e., the present manuscript. Rather, it is a concern that applies to Landgraf et al. 2021, where the experimental part is presented. I still do think that this is a serious concern. Replication is simply and inarguably a key principle of ecological and ecohydrological field research (not only, but especially those disciplines).

iii.), Dual-isotope space: I still believe that providing both stable water isotopes would greatly have benefited the study. Having that said, I do not expect the authors to incorporate it into this study as the manuscript is already complex and rich in information. The advantage of utilizing both isotopes (e.g. in dual-isotope plots) has been shown beneficial and used in hundreds of studies. It helps to understand evaporative conditions and interpret the degree of fractionation a particular sample/tree/species is subjected to by providing one simple measure. I also believe it would be highly beneficial in terms of model evaluation (measured vs. simulated lc-excess, for instance).

iv.) Complexity of the model: The model is, without doubt, incredibly powerful and for the first time I feel that all important features of the critical zone are represented sufficiently. It is also very cool to see the in situ isotope measurements do provide benefits for modeling purposes. Hence, it has great opportunities for future applications and I applaud the authors for the developing this highly useful model. However, I think it is also extremely data demanding and not simple to parameterize by any means. When applying the model, it needs to be carefully decided if cost and benefit are in balance. When reading the manuscript, I frequently asked myself: 'Are there optional components in the model, or is all this data always needed?' In other words, what are the minimum data requirements? Working in remote areas (mainly forests) since many years, I was wondering for instance, how rooting distributions would be incorporated in systems with more heterogeneous root systems and deeper soils. Can the rooting distributions be calibrated? I know this is not

necessary to answer in this manuscript, but there are practical questions me (and probably many other readers) might have.

v.) This is a thought and not a criticism: In isotope science, a few per mil can make a huge difference. With all the uncertainties involved into measuring (more so for in situ, but also for destructive sampling) and modeling (and that was the background of my question on uncertainty), I am honestly not sure, how well we can describe natural systems currently. The community is very active at present, but – in my opinion – we need to be critical and honest with ourselves in this regard. In the revised version, the authors have included a number of statements touching on this issue, which is much appreciated.

I congratulate all the authors for the study and wish them all the best for future applications.

Matthias Beyer

- Abstract, l.17-18: I think a core strength of the model is to provide E/T separations for different root plant types/ or plots (and that in a high temporal resolution). I think it would be interesting here to have the (overall) quantitative numbers for E/T or ET for willow and grass at the study site.
- L. 21: perhaps 'root water uptake' (and subsequently simply RWU) instead of 'root-uptake'? I think the former is more commonly used, but it is up to the authors to decide
- Introduction: I very much like the objectives as per R1
- L.165: 'unrealistically enriched isotopic compositions' – this is interesting. Is this known to be the case when wounding occurs? (this is a question out of interest, because I do not know)
- L.224/225: (suggestion to add) and no fractionation during root water uptake?
- L.280: 'estimate xylem $\delta 2H$' – is this a typo, or do the authors mean only $\delta 2H$ and not $\delta 18O$ is estimated? → 'estimated xylem isotope values'?
- L.345: I honestly do not know what '100,000 Latin Hypercube Sample parameter sets' are. But I am not an everyday modeler as well. I still believe that a very brief explanation would benefit the non-modeler reader.
- L.357: typo: the unit of soil moisture content should be m3/m3
- L.366-367: 'soil water isotopes' should read $\delta 2H$, I think? Because in table S2 I only see $\delta 2H$
- L.679: replace 'deuterium' by '$\delta 2H$ values'?

---

## Author Response (AR2)

**Response to Reviewer 1**

**General Comments**

I thank the authors for their revision of the manuscript, which satisfactorily address most of the comments. I find the paper clearer and even more enjoyable to read.

There are, however, a few minor points that still need clarification, as detailed below. After this, I consider this paper as fit for publication in HESS.

**Response to Reviewer 1 General Comments**

The authors thank the reviewer for their perceptive and helpful comments throughout the revision. The comments have greatly aided in improving the clarity and demonstrating the significant of the study.

**Specific Comments**

**R1C1:** L266-267: Contrary to what the authors state in the revised manuscript, and in their response to R1C16, the rooting depth in EcH2O is not set equal by default to total domain depth, at least in the root water uptake calculation one can find on http://bitbucket.igb-berlin.de:7990/users/ech2o/repos/ech2o_iso/browse/Hydro/SolveCanopyFluxes.cpp ; there, rooting depth it is the exact depth at which the cumulative fraction (from the surface) reaches 95%, see l. 248.

**Response to R1C1:** As the reviewer has shown, EcH2O calculates a rooting depth of 95% of the roots. However, this 95% is still based on using the whole domain for estimating total root growth. The relatively small impact of total rooting depth on the root fraction in different soil layers results in further negligible differences between rooting depth and total model domain for water use allocation. Lastly, for the calculations used in this manuscript (and inconsistency with the *kroot* distribution in EcH2O) it is most accurate for L266-267 to describe the use of $D$ as the total soil depth as that is what was used.

**R1C2:** L329-332: Maybe mention that "LAI calibration" refers to calibration Step 2 in the next paragraph.

**Response to R1C2:** The authors have included a reference to calibration step 2 for the LAI calibration,

**R1C3:** L341-347: I thank the authors for reworking this section, yet some aspects still need clarification: First, it is still not clear to me how the information from Step 1 was used in Step 2; were the 100,000 latin hypercube samples in Step 2 based on the parameter range from Step 1? Some other connection? ◦ It is quite surprising that in Step 2 none of the vegetation parameter related to allocation and growth were calibrated, contrary to earlier work by the same research group with the same model (Douinot et al., 2019). One expect that some degree of sensitivity for leaf area index and basal area simulations to these parameters. Can the authors justify this choice, perhaps using some sensitivity analysis or providing known literature values? ◦ Which "temporal dynamics of LAI" were used as inputs in Step 1? The same of as the calibration datasets of Step 2? ◦ How were the "infeasible parameter combination" assessed?

**Response to R1C3:** We have now clarified the calibration. Parameter sets (complete sets) were randomly resampled and merged with Latin hypercube sampling of vegetation biomass parameters. In doing so, soil parameter sets were not affected. We mistakenly left out the biomass parameters from the included

vegetation parameters in the previous revision. They have been included in the revised supplementary material. The authors have further clarified that MODIS LAI dynamics were used as inputs for calibration Step 1, and how infeasible parameter combinations were evaluated. (L341-347).

**R1C4:** L426: Puzzled by this somewhat strong allocation rule, I went on checking the grass allocation calculation in the available EcH2O-iso code (http://bitbucket.igb-berlin.de:7990/users/ech2o/repos/ech2o_iso/browse/Forest/GrowGrass.cpp) and the paper it draws from (Lozano-Parra et al., 2014). There, I could not find an obvious reason why the biomass allocation (total and root/leaf fractions) should be calculated as constant. Unless the LAI and/or NPP was kept constant in the simulations? Could the authors bring some precisions?

**Response to R1C4:** We thank the reviewer for identifying this error. Model output of the grass allocation was erroneously output as a single value, the authors have revised the statement and updated the table to reflect the seasonal cycle.

**R1C5:** Eq. (1): root proportions should all be in lower capitals.

**Response to R1C5:** Revised (Eq1).

**R1C6:** L357: "m3 m-3" instead of "m2/m2"

**Response to R1C6:** Revised (L357)

**R1C7:** Table S1: I am guessing the "aspect ratio" parameter the "root aspect ratio"? Please clarify.

**Response to R1C6:** Revised

**Response to Matthias Beyer**

**General Comments**
I thank the authors for providing a comprehensive and careful revised version of the manuscript that addresses many of the comments from version 1. I will first comment on the general points and then provide a few more comments that might help. Having that said, I recommend the manuscript for publication in HESS after addressing those very minor comments.

**Response to General Comments**
The authors thank Matthias for his insightful comments as these are important merit for future modelling in ecohydrology. The comments throughout the revision have been helpful in clarifying the manuscript and expanding the wider reach of the study.

**Specific Comments**
**R2C1:** Abstract, l.17-18: I think a core strength of the model is to provide E/T separations for different root plant types/ or plots (and that in a high temporal resolution). I think it would be interesting here to have the (overall) quantitative numbers for E/T or ET for willow and grass at the study site.

**Response to R2C1:** The authors have added some of the ET partitions to the abstract to improve the context of water use at the site. (L17-18).

**R2C2:** L. 21: perhaps 'root water uptake' (and subsequently simply RWU) instead of 'root-uptake'? I think the former is more commonly used, but it is up to the authors to decide

**Response to R2C2:** Revised (L21).

**R2C3:** Introduction: I very much like the objectives as per R1

**Response to R2C3:** Thank you.

**R2C4:** L.165: 'unrealistically enriched isotopic compositions' – this is interesting. Is this known to be the case when wounding occurs? (this is a question out of interest, because I do not know)

**Response to R2C4:** In our case the sudden marked enrichment and dissipation of enrichment (≫20 per mille above long term values) was coincided with the observed wounding and healing of the tree following installation. This may not be as severe for all installations or in all trees but needs further evaluation/exploration.

**R2C5:** L.224/225: (suggestion to add) and no fractionation during root water uptake?

**Response to R2C5:** Revised (L224)

**R2C6: L.280:** 'estimate xylem $\delta 2H$' – is this a typo, or do the authors mean only $\delta 2H$ and not $\delta 18O$ is estimated? → 'estimated xylem isotope values'?

**Response to R2C6:** Revised (L280)

**R2C7:** L.345: I honestly do not know what '100,000 Latin Hypercube Sample parameter sets' are. But I am not an everyday modeler as well. I still believe that a very brief explanation would benefit the non-modeler reader.

**Response to R2C7:** The authors have revised the statement to more clearly describe the parameter generation. The revision more clearly indicates that Latin Hypercube sampling is a method to acquire the parameter sets.

**R2C8:** L.357: typo: the unit of soil moisture content should be m3/m3

**Response to R2C8:** Revised (L357)

**R2C9:** L.366-367: 'soil water isotopes' should read $\delta 2H$, I think? Because in table S2 I only see $\delta 2H$

**Response to R2C9:** The authors have revised to indicate that the MAE presented is for $\delta 2H$. (L366)

**R2C10:** L.679: replace 'deuterium' by '$\delta 2H$ values'?

**Response to R2C10:** Revised. (L679)